# Graph Neural Networks with Adaptive Residual

**Xiaorui Liu**[1]
xiaorui@msu.edu

**Jiayuan Ding**[1]
dingjia5@msu.edu

**Wei Jin**[1]
jinwei2@msu.edu

**Han Xu**[1]
xuhan1@msu.edu

**Yao Ma**[2]
yao.ma@njit.edu

**Zitao Liu**[3]
liuzitao@100tal.com

**Jiliang Tang**[1]
tangjili@msu.edu

[1]Michigan State University, East Lansing, MI, USA
[2]New Jersey Institute of Technology, Newark, NJ, USA
[3]TAL Education Group, Beijing, China

## Abstract

Graph neural networks (GNNs) have shown the power in graph representation learning for numerous tasks. In this work, we discover an interesting phenomenon that although residual connections in the message passing of GNNs help improve the performance, they immensely amplify GNNs' vulnerability against abnormal node features. This is undesirable because in real-world applications, node features in graphs could often be abnormal such as being naturally noisy or adversarially manipulated. We analyze possible reasons to understand this phenomenon and aim to design GNNs with stronger resilience to abnormal features. Our understandings motivate us to propose and derive a simple, efficient, interpretable, and adaptive message passing scheme, leading to a novel GNN with Adaptive residual, AirGNN[1]. Extensive experiments under various abnormal feature scenarios demonstrate the effectiveness of the proposed algorithm.

## 1 Introduction

Recent years have witnessed the great success of graph neural networks (GNNs) in representation learning for graph structure data [1]. Essentially, GNNs generalize deep neural networks (DNNs) from regular grids, such as image, video and text, to irregular data such as social, energy, transportation, citation, and biological networks. Such data can be naturally represented as graphs with nodes and edges. The key building block for such generalization is the neural message passing framework [2]:

$$\mathbf{x}_u^{(k+1)} = \text{UPDATE}^{(k)}\big(\mathbf{x}_u^{(k)}, \mathbf{m}_{\mathcal{N}(u)}^{(k)}\big) \tag{1}$$

where $\mathbf{x}_u^{(k)} \in \mathbb{R}^d$ denotes the feature vector of node $u$ in the $k$-th iteration of message passing, and $\mathbf{m}_{\mathcal{N}(u)}^{(k)}$ is the message aggregated from $u$'s neighborhood $\mathcal{N}(u)$. The specific design of message passing scheme can be motivated from spectral domain [3, 4] or spatial domain [5, 6, 7, 2]. It usually linearly smooths the features in a local neighborhood on the graph.

GNNs have achieved superior performance in a large number of benchmark datasets [8] where the node features are assumed to be complete and informative. However, in real-world applications, some node features could be abnormal from various aspects. For instance, in social networks, new users might not have complete profile before they make connections with others, leading to missing user features. In transportation networks, node features can be noisy since there exist certain

---

[1]The implementation is available at `https://github.com/lxiaorui/AirGNN`.

uncertainty and dynamics in the observation of the traffic information. What is worse, node features can be adversarially chosen by the attacker to maliciously manipulate the prediction made by GNNs. Therefore, it is greatly desired to design GNN models with stronger resilience to abnormal node features.

In this work, we first perform empirical investigations on how representative GNN models behave on graphs with abnormal features. Specifically, based upon standard benchmark datasets, we simulate the abnormal features by replacing the features of randomly selected nodes with random Gaussian noise. Then the performance of node classification on abnormal features and normal features are examined separately. From our preliminary study in Section 2, we reveal two interesting observations: (1) Feature aggregation can boost the resilience to abnormal features, but too many aggregations could hurt the performance on both normal and abnormal features; and (2) Residual connection helps GNNs benefit from more layers for normal features, while making GNNs more fragile to abnormal features. We then provide possible explanations to understand these observed phenomena from the perspective of graph Laplacian smoothing. Our analyses imply that there might exist an intrinsic tension between feature aggregation and residual connection, which results in a performance tradeoff between normal features and abnormal features.

Motivated by these findings and understandings, we aim to design new GNNs with stronger resilience to abnormal features while largely maintaining the performance on normal features. Our contributions can be summarized as follows:

- We discover an intrinsic tension between feature aggregation and residual connection in GNNs, and the corresponding performance tradeoff between abnormal and normal features. We also analyze possible reasons to explain and understand these findings.
- We propose a simple, efficient, principled and adaptive message passing scheme, which leads to a novel GNN model with adaptive residual, named as AirGNN.
- Extensive experiments under various abnormal feature scenarios demonstrate the superiority of the proposed algorithm. The ablation study demonstrates how the adaptive residuals mitigate the impact of abnormal features.

## 2 Preliminary

Before introducing the preliminary study, we first define the notations used throughout the paper.

**Notations.** We use bold upper-case letters such as $\mathbf{X}$ to denote matrices. Given a matrix $\mathbf{X} \in \mathbb{R}^{n \times d}$, we use $\mathbf{X}_i$ to denote its $i$-th row and $\mathbf{X}_{ij}$ to denote its element in $i$-th row and $j$-th column. The Frobenius norm and $\ell_{21}$ norm of a matrix $\mathbf{X}$ are defined as $\|\mathbf{X}\|_F = \sqrt{\sum_{ij} \mathbf{X}_{ij}^2}$ and $\|\mathbf{X}\|_{21} = \sum_i \|\mathbf{X}_i\|_2 = \sum_i \sqrt{\sum_j \mathbf{X}_{ij}^2}$, respectively. We define $\|\mathbf{X}\|_2 = \sigma_{\max}(\mathbf{X})$ where $\sigma_{\max}(\mathbf{X})$ is the largest singular value of $\mathbf{X}$.

Let $\mathcal{G} = \{\mathcal{V}, \mathcal{E}\}$ be a graph with the node set $\mathcal{V} = \{v_1, \ldots, v_n\}$ and the undirected edge set $\mathcal{E} = \{e_1, \ldots, e_m\}$. We use $\mathcal{N}(v_i)$ to denote the neighboring nodes of node $v_i$, including $v_i$ itself. Suppose that each node is associated with a $d$-dimensional feature vector, and the features for all nodes are denoted as $\mathbf{X}_{\text{fea}} \in \mathbb{R}^{n \times d}$. The graph structure $\mathcal{G}$ can be represented as an adjacent matrix $\mathbf{A} \in \mathbb{R}^{n \times n}$, where $\mathbf{A}_{ij} = 1$ when there exists an edge between nodes $v_i$ and $v_j$, and $\mathbf{A}_{ij} = 0$ otherwise. The graph Laplacian matrix is defined as $\mathbf{L} = \mathbf{D} - \mathbf{A}$, where $\mathbf{D}$ is the diagonal degree matrix. Let us denote the commonly used feature aggregation matrix in GNNs [3] as $\tilde{\mathbf{A}} = \hat{\mathbf{D}}^{-\frac{1}{2}} \hat{\mathbf{A}} \hat{\mathbf{D}}^{-\frac{1}{2}}$ where $\hat{\mathbf{A}} = \mathbf{A} + \mathbf{I}$ is the adjacent matrix with self-loop and its degree matrix is $\hat{\mathbf{D}}$. The corresponding Laplacian matrix is defined as $\tilde{\mathbf{L}} = \mathbf{I} - \tilde{\mathbf{A}}$.

In this work, we focus on the setting where a subset of nodes in the graph contain abnormal features, while the remaining nodes have normal features. In the remaining of this paper, we use *abnormal/normal features* to denote *nodes with abnormal/normal features*, for simplicity.

### 2.1 Preliminary Study

**Experimental setup.** To investigate how GNNs behave on abnormal and normal node features, we design semi-supervised node classification experiments on three common datasets (i.e., Cora,

CiteSeer and PubMed), following the data splits in the work [3]. Moreover, we simulate the abnormal features by assigning 10% of the nodes with random features sampled from a standard Gaussian distribution. The experiments are performed on representative GNN models covering coupled and decoupled architectures, including GCN [3], GCNII [9], APPNP [10], and their variants with or without residual connections in feature aggregations, denoted as w/Res and wo/Res. All methods follow the hyperparameter settings in their original papers. We examine how these models perform when the number of layers increases. Note that for the decoupled architectures such as APPNP, we fix the 2-layer MLP and increase the number of propagation layers. While for the coupled architectures such as GCN and GCNII, we increase the number of feature transformation and propagation layers simultaneously. We report the average performance over 10 times of random selection of the noise node sets. The node classification accuracy (mean and standard variance) on nodes with abnormal and normal features is illustrated in Figure 1 and Figure. 2, separately.

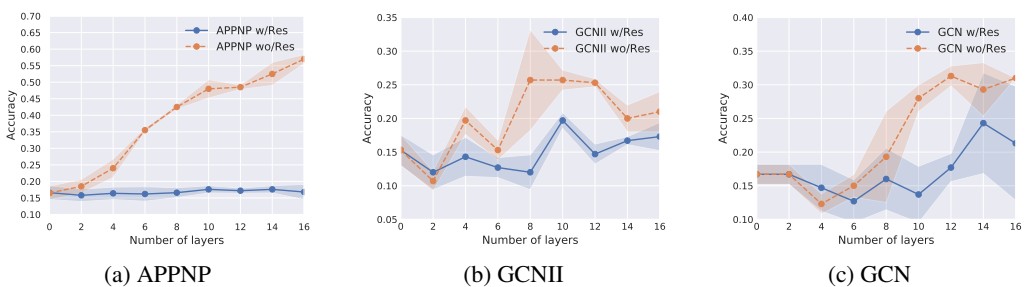

Figure 1: Node classification accuracy on abnormal nodes (Cora)

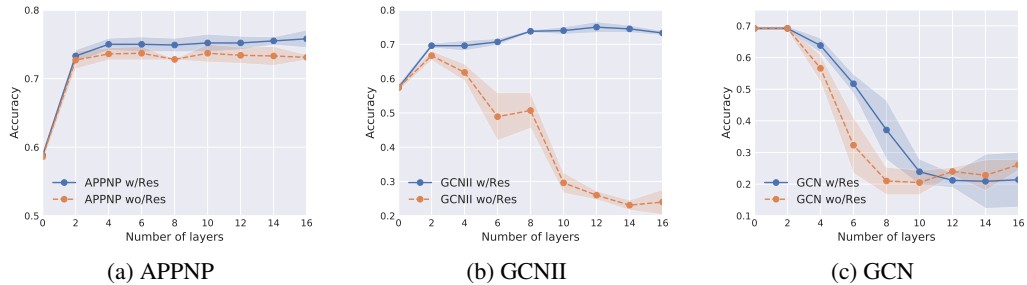

Figure 2: Node classification accuracy on normal nodes (Cora)

**Observations.** From Figure 1 and Figure 2, we can make the following observations: (1) Without residual connection, more layers (e.g., $> 2$ for GCN and GCNII, $> 10$ for APPNP) hurt the accuracy on nodes with normal features. However, more layers boost the accuracy on nodes with abnormal features significantly, before finally starting to decrease; (2) With residual connection, the accuracy on nodes with normal features keeps increasing with more layers[2]. However, the accuracy on nodes with abnormal features only increases marginally when stacking more layers, and then starts to decrease. While we only present the experiments on Cora, we defer the results on other datasets to Appendix C, which provide similar observations. To conclude, we can summarize these observations into two major findings:

- *Finding I:* Feature aggregation can boost the resilience to abnormal features, but too many aggregations could hurt the performance on both normal and abnormal nodes;

- *Finding II:* Residual connection helps GNNs benefit from more layers for nodes with normal features, while making GNNs more fragile to abnormal features.

---

[2]GCN w/Res is an exception because its residual is not appropriate, which is consistent with the experiments in the work [3].

## 2.2 Understandings

In this subsection, we provide the understanding and explanation for aforementioned findings, from the perspective of graph Laplacian smoothing.

**Understanding Finding I: Feature aggregation as Laplacian smoothing**

The message passing in GCN [3], GCNII wo/ residual and APPNP wo/ residual (as well as many popular GNN models), follows the feature aggregation

$$\mathbf{X}_{\text{out}} = \tilde{\mathbf{A}}\mathbf{X}_{\text{in}}, \tag{2}$$

where $\mathbf{X}_{\text{in}}$ and $\mathbf{X}_{\text{out}}$ represent the features before and after message passing layer, respectively. It can be interpreted as one gradient descent step for the Laplacian smoothing problem [11]

$$\underset{\mathbf{X}\in\mathbb{R}^{n\times d}}{\arg\min}\ \mathcal{L}_1(\mathbf{X}) := \frac{1}{2}\text{tr}\Big(\mathbf{X}^\top(\mathbf{I}-\tilde{\mathbf{A}})\mathbf{X}\Big) = \frac{1}{2}\sum_{(v_i,v_j)\in\mathcal{E}}\|\frac{\mathbf{X}_i}{\sqrt{d_i+1}} - \frac{\mathbf{X}_j}{\sqrt{d_j+1}}\|_2^2, \tag{3}$$

where $d_i$ is the node degree of node $v_i$. Eq. (2) can be derived from $\mathbf{X}_{\text{out}} = \mathbf{X}_{\text{in}} - (\mathbf{I}-\tilde{\mathbf{A}})\mathbf{X}_{\text{in}} = \tilde{\mathbf{A}}\mathbf{X}_{\text{in}}$, with the initialization $\mathbf{X} = \mathbf{X}_{\text{in}}$ and stepsize $\gamma = 1$. The Laplacian smoothing problem penalizes the feature difference between neighboring nodes. To reduce this penalty, the feature aggregation in Eq. (2) smooths the node features by taking the average of local neighbors, and thus can be considered as low-pass filter which gradually filters out high-frequency signals [12, 13]. Therefore, it increases the resilience to abnormal features which are likely to be high-frequency signals. In other words, the local neighboring nodes help to correct the abnormal features. Unfortunately, if applied too many times, these low-pass filters could overly smooth the features (well-known as oversmoothing [14, 15]) such that nodes are not distinguishable enough, providing an explanation to the degraded performance on both abnormal and normal features when stacking too many layers.

**Understanding Finding II: Residual connection maintains feature proximity**

To adjust the feature smoothness for better performance, APPNP [10] utilizes residual connections in message passing as follows

$$\mathbf{X}^{k+1} = (1-\alpha)\tilde{\mathbf{A}}\mathbf{X}^k + \alpha\mathbf{X}_{\text{in}}, \tag{4}$$

where $\mathbf{X}^0 = \mathbf{X}_{\text{in}}$. It can be considered as an iterative solution for the regularized Laplacian smoothing problem [11]

$$\underset{\mathbf{X}\in\mathbb{R}^{n\times d}}{\arg\min}\ \mathcal{L}_2(\mathbf{X}) := \frac{\alpha}{2(1-\alpha)}\|\mathbf{X}-\mathbf{X}_{\text{in}}\|_F^2 + \frac{1}{2}\text{tr}\Big(\mathbf{X}^\top(\mathbf{I}-\tilde{\mathbf{A}})\mathbf{X}\Big), \tag{5}$$

with initialization $\mathbf{X} = \mathbf{X}_{\text{in}}$ and stepsize $\gamma = 1-\alpha$ due to

$$\mathbf{X}^{k+1} = \mathbf{X}^k - (1-\alpha)\Big(\frac{\alpha}{1-\alpha}(\mathbf{X}^k - \mathbf{X}_{\text{in}}) + (\mathbf{I}-\tilde{\mathbf{A}})\mathbf{X}^k\Big) = (1-\alpha)\tilde{\mathbf{A}}\mathbf{X}^k + \alpha\mathbf{X}_{\text{in}}.$$

GCNII [9] adopts a similar message passing but further combines a feature transformation layer in each message passing step, which leads to a coupled architecture, as contrast to the decoupled architecture of APPNP. The residual connection naturally arises when regularizing the proximity between input and output features, as showed in the first term of $\mathcal{L}_2(\mathbf{X})$. Such proximity can help avoid the trivial solution for the problem in Eq. (3), i.e., totally oversmoothed features only depending on node degrees, and consequently mitigates the oversmoothing issue. More intuitively, residual connections in GNNs provide direct information flows between layers that can preserve some necessary high-frequency signals for better discrimination between classes. More layers with residual provide a more accurate solution to Eq. (5), which explains the performance gain from deeper GNNs. Unfortunately, these residual connections also undesirably carry on abnormal features which are detrimental, leading to the inferior performance on abnormal features.

## 3 The Proposed Framework

In this section, we first motivate the proposed adaptive message passing scheme (AMP) with further discussions on our preliminary study. We then introduce more details about AMP, its interpretations, convergence guarantee and computation complexity, as well as the model architecture of AirGNN.

### 3.1 Design Motivation

Our preliminary study in Section 2 reveals an intrinsic tension between feature aggregation and residual connection: (1) feature aggregation helps smooth out abnormal features, while it could cause inappropriate smoothing for normal features; (2) residual connection is essential for adjusting the feature smoothness, but it could be detrimental for abnormal features. Although this conflict can be partially mitigated by adjusting the residual connection such as the residual weight $\alpha$ in GCNII [9] and APPNP [10], such global adjustment cannot be adaptive to a subset of the nodes, e.g., the nodes with abnormal features. This is crucial because in practice we often encounter the scenario where only a subset of nodes contain abnormal features. Therefore, how to reconcile this dilemma still desires dedicated efforts. We then naturally ask a question: *Can we design a better message passing scheme with node-wise adaptive feature aggregation and residual connection?*

The motivation of the proposed idea builds upon the following intuition: while it is important to maintain the proximity between input and output features as in Eq. (5), it could be over aggressive to penalize their deviations by the square of Frobenius norm, i.e., $\|\mathbf{X} - \mathbf{X}_{\text{in}}\|_F^2 = \sum_{i=1}^{n} \|\mathbf{X}_i - (\mathbf{X}_{\text{in}})_i\|_2^2$. The fact that this penalty does not tolerate large deviations weakens the capability to remove abnormal features through Laplacian smoothing. This motivates us to consider an alternative proximity penalty

$$\|\mathbf{X} - \mathbf{X}_{\text{in}}\|_{21} := \sum_{i=1}^{n} \|\mathbf{X}_i - (\mathbf{X}_{\text{in}})_i\|_2, \tag{6}$$

which instead penalizes the deviations by the $\ell_1$ norm of row-wise $\ell_2$ norms, namely $\ell_{21}$ norm. The $\ell_{21}$ norm promotes row sparsity in $\mathbf{X} - \mathbf{X}_{\text{in}}$, and it also allows large deviations because the penalty on large values is less aggressive, leading to the potential removal of abnormal features. Therefore, we propose the following Laplacian smoothing problem regularized by $\ell_{21}$ norm proximity control:

$$\underset{\mathbf{X} \in \mathbb{R}^{n \times d}}{\arg\min} \lambda \|\mathbf{X} - \mathbf{X}_{\text{in}}\|_{21} + \frac{1}{2} \text{tr}(\mathbf{X}^\top (\mathbf{I} - \tilde{\mathbf{A}})\mathbf{X}), \tag{7}$$

where $\lambda \in [0, \infty)$ is a parameter to adjust the balance between proximity and Laplacian smoothing. In order to easy the tuning of $\lambda$, we made a modification of Eq. (7):

$$\underset{\mathbf{X} \in \mathbb{R}^{n \times d}}{\arg\min} \ \mathcal{L}(\mathbf{X}) := \lambda \|\mathbf{X} - \mathbf{X}_{\text{in}}\|_{21} + (1 - \lambda) \text{tr}(\mathbf{X}^\top (\mathbf{I} - \tilde{\mathbf{A}})\mathbf{X}), \tag{8}$$

where $\lambda \in [0, 1]$ controls the balance.

### 3.2 Adaptive Message Passing

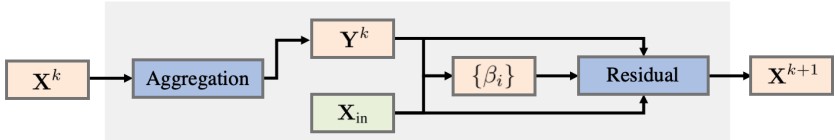

Figure 3: Diagram of Adaptive Message Passing

$\mathcal{L}(\mathbf{X})$ is a composite objective with non-smooth and smooth components. We optimize it by proximal gradient descent [16] and obtain the following iterations as the adaptive message passing (AMP):

$$\mathbf{Y}^k = \mathbf{X}^k - 2\gamma(1 - \lambda)(\mathbf{I} - \tilde{\mathbf{A}})\mathbf{X}^k = \big(1 - 2\gamma(1 - \lambda)\big)\mathbf{X}^k + 2\gamma(1 - \lambda)\tilde{\mathbf{A}}\mathbf{X}^k \tag{9}$$

$$\mathbf{X}^{k+1} = \underset{\mathbf{X}}{\arg\min} \left\{ \lambda \|\mathbf{X} - \mathbf{X}_{\text{in}}\|_{21} + \frac{1}{2\gamma} \|\mathbf{X} - \mathbf{Y}^k\|_F^2 \right\} \tag{10}$$

where $\mathbf{X}^0 = \mathbf{X}_{\text{in}}$ and $\gamma$ is the stepsize to be specified later. Let $\mathbf{Z} = \mathbf{X} - \mathbf{X}_{\text{in}}$, and Eq. (10) can be rewritten as:

$$\mathbf{Z}^{k+1} = \underset{\mathbf{Z}}{\arg\min} \left\{ \lambda \|\mathbf{Z}\|_{21} + \frac{1}{2\gamma} \|\mathbf{Z} - (\mathbf{Y}^k - \mathbf{X}_{\text{in}})\|_F^2 \right\}$$

$$= \mathbf{prox}_{\gamma\lambda\|\cdot\|_{21}}(\mathbf{Y}^k - \mathbf{X}_{\text{in}}) \tag{11}$$

$$\mathbf{X}^{k+1} = \mathbf{X}_{\text{in}} + \mathbf{Z}^{k+1}. \tag{12}$$

The $i$-th row of the proximal operator in Eq. (11) can be computed analytically

$$\left(\mathbf{prox}_{\gamma\lambda\|\cdot\|_{21}}(\mathbf{X})\right)_i = \frac{\mathbf{X}_i}{\|\mathbf{X}_i\|_2}\max(\|\mathbf{X}_i\|_2 - \gamma\lambda, 0) = \max(1 - \frac{\gamma\lambda}{\|\mathbf{X}_i\|_2}, 0)\cdot\mathbf{X}_i. \quad (13)$$

Note that the proximal operator returns $\mathbf{0}$ if the input vector is $\mathbf{0}$. Substituting $\mathbf{X}$ in Eq. (13) with $\mathbf{Y}^k - \mathbf{X}_{\text{in}}$ and combining Eq. (11) and Eq. (12), then Eq. (12) becomes

$$\mathbf{X}_i^{k+1} = (\mathbf{X}_{\text{in}})_i + \beta_i(\mathbf{Y}_i^k - (\mathbf{X}_{\text{in}})_i) = (1 - \beta_i)(\mathbf{X}_{\text{in}})_i + \beta_i\mathbf{Y}_i^k, \quad \forall i \in [n], \quad (14)$$

where $\beta_i := \max(1 - \frac{\gamma\lambda}{\|\mathbf{Y}_i^k - (\mathbf{X}_{\text{in}})_i\|_2}, 0)$. To summarize, the proposed adaptive message passing (AMP) scheme is showed in Figure 4, and a diagram is showed in Figure 3. In detail, AMP works as follows:

- The first step takes a *feature aggregation* within the local neighbors with a self-loop weighted by $1 - 2\gamma(1 - \lambda)$;
- The second step computes a weight $\beta_i \in [0, 1]$ for each node $v_i$ depending on the local deviation $\|\mathbf{Y}_i^k - (\mathbf{X}_{\text{in}})_i\|_2$.
- The final step takes a *linear combination* of input features $\mathbf{X}_{\text{in}}$ and the aggregated features $\mathbf{Y}^k$, where the node-wise residual is adaptively weighted by $1 - \beta_i$ for each node $v_i$.

$$\begin{cases} \mathbf{Y}^k &= \left(1 - 2\gamma(1 - \lambda)\right)\mathbf{X}^k + 2\gamma(1 - \lambda)\tilde{\mathbf{A}}\mathbf{X}^k \\ \beta_i &= \max(1 - \dfrac{\gamma\lambda}{\|\mathbf{Y}_i^k - (\mathbf{X}_{\text{in}})_i\|_2}, 0) \quad \forall i \in [n] \\ \mathbf{X}_i^{k+1} &= (1 - \beta_i)(\mathbf{X}_{\text{in}})_i + \beta_i\mathbf{Y}_i^k \qquad \forall i \in [n] \end{cases}$$

Figure 4: Adaptive Message Passing (AMP)

The convergence guarantee of AMP and parameter setting for the stepsize $\gamma$ are illustrated in Theorem 1 and proved in Appendix A. According to Theorem 1, if we set $\gamma = \frac{1}{4(1-\lambda)}$ or $\gamma = \frac{1}{2(1-\lambda)}$, then the first step of AMP can be simplified as $\mathbf{Y}^k = \frac{1}{2}\mathbf{X}^k + \frac{1}{2}\tilde{\mathbf{A}}\mathbf{X}^k$ and $\mathbf{Y}^k = \tilde{\mathbf{A}}\mathbf{X}^k$, respectively. The choice of stepsize will only impact the convergence speed but not the ultimate effect of AMP when it convergences to the fixed point solution. We also discuss the computation complexity per iteration of AMP in Remark 1.

**Theorem 1** (Convergence of AMP). *Under the stepsize setting $\gamma < \frac{1}{(1-\lambda)\|\tilde{\mathbf{L}}\|_2}$, the proposed adaptive message passing scheme (AMP) in Eq. (9) and Eq. (10) converges to the optimal solution of the problem defined in Eq. (8). In practice, it is sufficient to choose any $\gamma < \frac{1}{2(1-\lambda)}$ since $\|\tilde{\mathbf{L}}\|_2 \leq 2$. Moreover, if the connected components of the graph $\mathcal{G}$ are not bipartite graphs, it is sufficient to choose $\gamma = \frac{1}{2(1-\lambda)}$ since $\|\tilde{\mathbf{L}}\|_2 < 2$.*

**Remark 1** (Computation complexity). *AMP is as efficient as simple feature aggregation $\mathbf{X}_{out} = \tilde{\mathbf{A}}\mathbf{X}_{in}$ because the additional computation cost from the second and third steps in Figure 4 is in the order $\mathcal{O}(nd)$, where $n$ is the number of nodes and $d$ is the feature dimension. This is negligible compared with the computation cost $\mathcal{O}(md)$ in feature aggregation, where $m$ is the number of edges, due to the fact that usually there are many more edges than nodes in real-world graphs, i.e., $m \gg n$.*

### 3.3 Interpretation of AMP

Interestingly, the proposed AMP has a simple and intuitive interpretation as adaptive residual connection, which aligns well with our design motivation:

- If the feature of node $v_i$, i.e., $(\mathbf{X}_{\text{in}})_i$, is significantly inconsistent with its local neighbors, i.e., the aggregated feature $\mathbf{Y}_i^k$, then the local deviation $\|\mathbf{Y}_i^k - (\mathbf{X}_{\text{in}})_i\|_2$ will be large, which leads to a $\beta_i$ close to 1. Therefore, the final step will assign a small weight to the residual, i.e., $(1 - \beta_i)(\mathbf{X}_{\text{in}})_i$, and the aggregated feature $\mathbf{Y}_i^k$ will dominate.

- On the contrary, if $(\mathbf{X}_{\text{in}})_i$ is already consistent with its local neighbors, $\|\mathbf{Y}_i^k - (\mathbf{X}_{\text{in}})_i\|_2$ will be small, which leads to a $\beta_i$ close to 0. Thus, the residual will dominate, which is reasonable since there is less need to aggregate features in this case.

- To summarize, the local deviation $\|\mathbf{Y}_i^k - (\mathbf{X}_{\text{in}})_i\|_2$ provides a natural transition from $\beta_i \to 1$ to $\beta_i \to 0$, and the transition can be modulated by $\lambda$ which can be either learned or tuned as a hyperparameter through cross-validation. This transition provides an node-wise adaptive residual connection for the message passing scheme.

**Adaptivity for abnormal & normal features.** According to the homophily assumption on graph structure data [17, 18, 19, 3], the feature representations of normal features should be more consistent with local neighbors than abnormal features. As a result, AMP will assign more residual (i.e., smaller $\beta$) to normal features but less residual (i.e., larger $\beta$) to abnormal features, providing a customized tradeoff between feature aggregation and residual connection. Consequently, it can promote both the resilience to abnormal features and the performance on normal features. Above discussion also implies a clear physical meaning for $\beta$ in AMP, and we formally define it as the adaptive score.

**Definition 1** (Adaptive score). *The variables $\{\beta_1, \cdots, \beta_n\}$ in the adaptive message passing scheme (AMP) are defined as the adaptive scores for nodes $\{v_1, \cdots, v_n\}$ respectively in graph $\mathcal{G}$. In particular, the larger $\beta_i$ is, the more likely the feature of node $v_i$ is abnormal.*

**Remark 2** (Nonlinear smoother). *Different from most existing message passing scheme which are linear smoothers, AMP is a nonlinear smoother because the weights $\{\beta_i\}$ are computed from $\mathbf{Y}^k$ and $\mathbf{X}_{in}$. This nonlinearity is the key to achieve adaptive residual connection for different nodes.*

### 3.4 The Model Architecture

The proposed adaptive message passing (AMP) can be used as a building block in many GNN models to improve the resilience to abnormal node features. In this work, we choose the the decoupled architectures as APPNP [10] and DAGNN [20], and propose the Adaptive residual GNN (AirGNN):

$$\mathbf{X}_{\text{in}} = h_\theta(\mathbf{X}_{\text{fea}}), \tag{15}$$

$$\mathbf{Y}_{\text{pre}} = \mathbf{AMP}\left(\mathbf{X}_{\text{in}}, K, \lambda\right). \tag{16}$$

$h_\theta(\cdot)$ is any machine learning model parameterized by learnable parameters $\theta$, such as multilayer perceptrons (MLPs). $\mathbf{X}_{\text{fea}} \in \mathbb{R}^{n \times d}$ denotes the initial node features. The model $h_\theta(\cdot)$ will first transform the initial node features as $\mathbf{X}_{\text{in}} = h_\theta(\mathbf{X}_{\text{fea}})$. AMP takes $h_\theta(\mathbf{X}_{\text{fea}})$ as input, and performs $K$ steps of AMP with the hyperparameter $\lambda$. Similar to the majority of existing GNN models, the training objective is the cross-entropy classification loss on the labeled nodes, and the whole model is trained in an end-to-end way. Note that AirGNN is very efficient as explained in Remark 1, and it only requires two hyperparameters $K$ and $\lambda$ without introducing additional parameters to learn, which could reduce the risk of overfitting.

## 4 Experiment

In this section, we aim to verify the effective of the proposed adaptive message passing scheme (AMP) and the AirGNN model through the semi-supervised node classification tasks. Specifically, we try to answer the following questions: (1) How does AirGNN perform on abnormal and normal features? (Section 4.2 and 4.3) and (2) How does AirGNN work by adjusting the adaptive residual? (Section 4.4)

### 4.1 Experimental Settings

**Datasets and baselines.** We conduct experiments on 8 real-world datasets including three citation graphs, i.e., Cora, Citeseer, Pubmed [21], two co-authorship graphs, i.e., Coauthor CS and Coauthor Physics [22], two co-purchase graphs, i.e., Amazon Computers and Amazon Photo [22], and one OGB dataset, i.e., ogbn-arxiv [23]. Due to the space limit, we only present the results on Cora, Citeseer, and Pubmed in this section, but defer the results on other datasets to Appendix D.1. More details about the data statistics and data splits are summarized in Appendix B. The proposed AirGNN is compared with representative GNNs, including GCN [3], GAT [6], APPNP [10] and GCNII [9]. We defer the comparison with the variants of APPNP and Robust GCN [24] to Appendix D.3 and D.4 respectively.

**Parameter settings.** For all baselines, we follow the best hyperparameter settings in their original papers. Additionally, we tune a best residual weight $\alpha$ for APPNP and GCNII in the range $[0, 1]$. For AirGNN, we use a two-layer MLP as the base model $h_\theta(\cdot)$, following APPNP. We fix the learning rate 0.01, dropout 0.8, and weight decay 0.0005. Moreover, we set $\gamma = \frac{1}{2(1-\lambda)}$ as suggested by Theorem 1. We choose $K = 10$ and tune $\lambda$ in the range $[0, 1]$. Adam optimizer [25] is used in all experiments. We run all experiments by 10 times, and report the mean and variance.

**Evaluation setting.** We assess the performance of all models under two types of abnormal feature scenarios, including noisy features and adversarial features. The abnormal features are injected to randomly selected test nodes after model training. By default, all hyperparameters are tuned according to the performance on validation sets when the dataset is clean. If tuning the hyperparameter $\lambda$ of AirGNN according to the validation sets after injecting abnormal features, the performance will be even better, as discussed in Appendix D.2. The performance on clean data are showed in Appendix D.5 to demonstrate that AirGNN doesn't need to sacrifice accuracy for better robustness against abnormal features.

### 4.2 Performance Comparison with Noisy Features

In this subsection, we consider the abnormal features in the noisy feature scenario. Specifically, we simulate the noisy features by assigning a subset of the nodes with random features sampled from a multivariate standard Gaussian distribution. Note that the selection of noise subsets has a apparent impact on the performance since some nodes are less vulnerable to abnormal features while others are more vulnerable. To reduce such variance, we report the average performance over 10 times of random selection of the noise node sets, similar to the settings in the preliminary study in Section 2. We report the node classification test accuracy on abnormal (noisy) features and normal features in Figure 5 and Figure 6, separately, under varying noisy ratio. From these figures, we can observe:

- Figure 5 shows that AirGNN significantly outperforms all baselines on all datasets in terms of the performance on noisy nodes. This verifies that AMP is able to improve the resilience to noisy features, aligning well with the design motivation.

- Figure 6 shows that AirGNN promotes the performance on normal nodes when abnormal nodes exist. This is because AMP can remove some abnormal features which are detrimental to normal nodes.

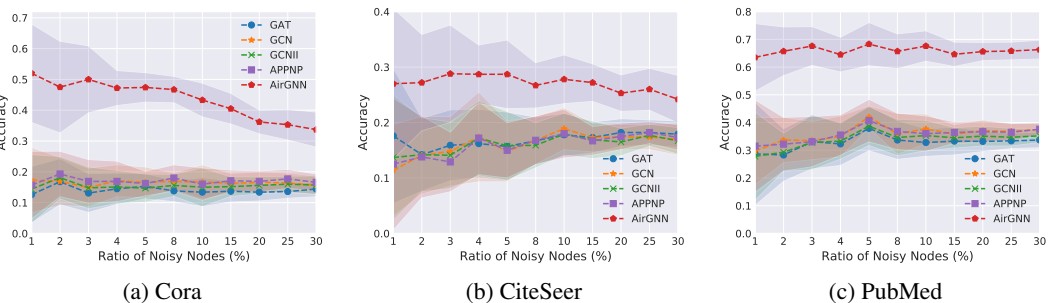

|  (a) Cora | (b) CiteSeer | (c) PubMed |

Figure 5: Node classification accuracy on abnormal (noisy) nodes

### 4.3 Performance Comparison with Adversarial Features

In this subsection, we consider the abnormal feature scenario when the node features are maliciously attacked by the attacker to manipulate the prediction of GNNs. We use the Nettack [26] implemented in DeepRobust[3] [27], a PyTorch library for adversarial attacks and defenses, to generate the adversarial features. We randomly choose 40 test nodes as the targeted nodes, and assess the performance under increasing perturbation budgets $\{0, 5, 10, 20, 50, 80\}$, where the perturbation numbers denote the number of feature dimensions that can be manipulated. The node classification accuracy on these attacked nodes are showed in Figure 7. From these figures, we can make the following observations:

---

[3]`https://github.com/DSE-MSU/DeepRobust`

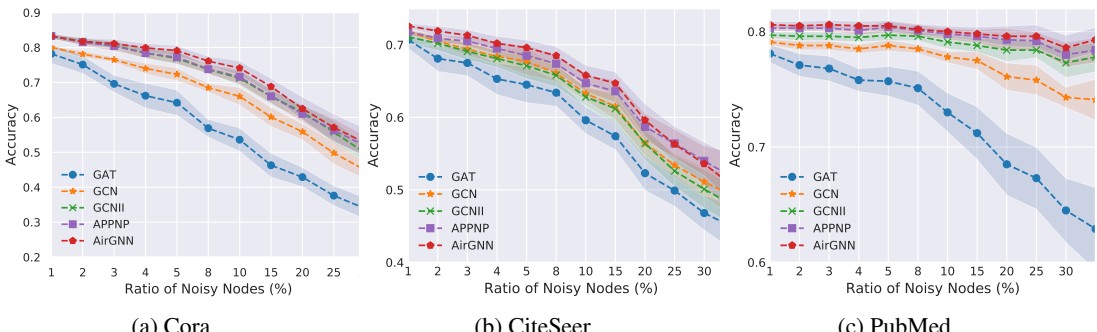

Figure 6: Node classification accuracy on normal nodes

- AirGNN is significantly more robust against adversarially attacked features than all baselines. MLP is the most vulnerable model, which demonstrates the usefulness of graph structure information in combating against abnormal node features.

- The advantages of AirGNN over the baselines become much stronger with larger perturbation budgets. This suggests that AMP can significantly improve the resilience to abnormal features.

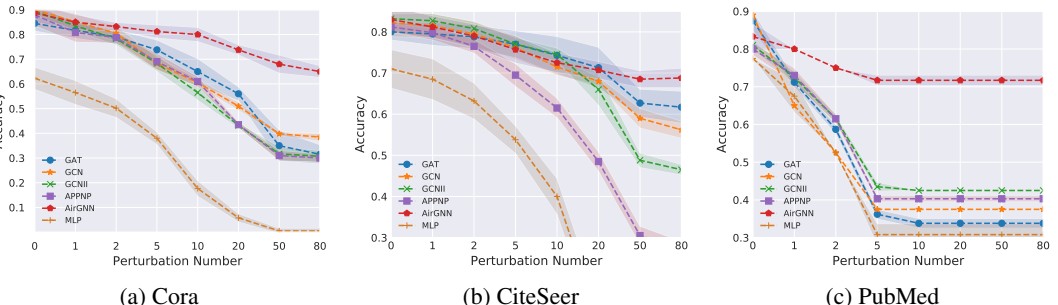

Figure 7: Node classification accuracy on adversarial nodes

### 4.4 Adaptive Residual for Abnormal & Normal Nodes

To further understand and verify how AMP and AirGNN work, we investigate the adaptive score $\beta_i$ for each node $v_i$. Specifically, the average adaptive scores for abnormal nodes and normal nodes in the last layer of AMP are computed separately. In the noisy feature scenario, we fix ratio of noisy nodes as 10%. In the adversarial feature scenario, we choose 40 target nodes and fix the perturbation number as 80. The results in noisy and adversarial feature scenarios are showed in Table 1 and Table 2, respectively. From these tables, we can observe:

- On the one hand, it can be clearly observed that in both scenarios, the average adaptive scores for abnormal nodes are significantly higher than those for normal nodes. Therefore, it verifies our intuition that large adaptive scores are strongly related to abnormal features.

- On the other hand, it also implies that the residual weights (i.e., $1 - \beta_i$) for abnormal nodes are much lower than those of normal nodes. This perfectly aligns with our motivation to remove abnormal features by reducing their residual connections.

The study on adaptive scores verifies how the adaptive residuals in AMP and AirGNN work as designed. It corroborates that AirGNN not only tremendously boosts the resilience to abnormal features but also provides interpretable information for anomaly detection that will be useful in many security-critical scenarios since the adaptive score serves as a good indicator of abnormal nodes. Morever, it is expected that APPNP without residual will perform well on abnormal nodes but it will sacrifice the performance on normal nodes. We provide detailed comparison with APPNP w/Res and APPNP wo/Res in Appendix D.3 to show the advantages of adaptive residual of AirGNN.

Table 1: Average adaptive score ($\beta$) and residual weight ($1 - \beta$) in the noisy feature scenario.

| Measure | Cora | CiteSeer | PubMed |
|---|---|---|---|
| Average adaptive score for abnormal nodes | $0.998 \pm 0.000$ | $0.988 \pm 0.000$ | $0.996 \pm 0.000$ |
| Average adaptive score for normal nodes | $0.924 \pm 0.002$ | $0.807 \pm 0.005$ | $0.869 \pm 0.006$ |
| Average residual weight for abnormal nodes | $0.002 \pm 0.000$ | $0.012 \pm 0.000$ | $0.004 \pm 0.000$ |
| Average residual weight for normal nodes | $0.076 \pm 0.002$ | $0.193 \pm 0.005$ | $0.131 \pm 0.006$ |

Table 2: Average adaptive score ($\beta$) and residual weight ($1 - \beta$) in the adversarial feature scenario.

| Measure | Cora | CiteSeer | PubMed |
|---|---|---|---|
| Average adaptive score for abnormal nodes | $0.987 \pm 0.000$ | $0.930 \pm 0.007$ | $0.959 \pm 0.005$ |
| Average adaptive score for normal nodes | $0.922 \pm 0.004$ | $0.689 \pm 0.024$ | $0.826 \pm 0.016$ |
| Average residual weight for abnormal nodes | $0.013 \pm 0.000$ | $0.070 \pm 0.007$ | $0.041 \pm 0.005$ |
| Average residual weight for normal nodes | $0.078 \pm 0.004$ | $0.311 \pm 0.024$ | $0.174 \pm 0.016$ |

## 5 Related Work

GNNs generalize convolutional neural networks (CNN) to graph structure data through the message passing framework [1, 2, 7]. The design of message passing and GNN architectures are majorly motivated in spectral domain [3, 4] and spatial domain [5, 6, 7, 2]. Recent works have shown that the message passing in GNNs can be regarded as low-pass graph filters [12, 13]. More generally, it has been proven that message passing in many GNNs can be uniformly derived from graph signal denoising [11, 28, 29, 30]. Classic GNNs such as GCN [3] and GAT [6] achieve their best performance with shallow models, but their performance degrades when stacking more layers, which can be partially explained through oversmoothing analyses [14, 15]. Recent works propose to use residual connections or skip connections to mitigate the oversmoothing issues, and they demonstrate the potential benefits from more feature aggregations. Examples include but not limited to DeepGCNs [31], JKNet [32], GCNII [9], APPNP [10] and DeeperGNN [20]. These models use global residual connection that can not be adaptive for each node, which significantly differ from the proposed AirGNN.

Recently, there are growing interests in reducing GNNs' vulnerability to the graph structure noise, such as Robust GCN [24], GCN-SVD [33], Pro-GNN [34], IDGL [35], ElasticGNN [36], etc. Please refer to the comprehensive surveys [37, 38] for more details. However, how to design GNNs with strong resilience to abnormal node features remains to be developed. To the best of our knowledge, AirGNN is the first GNN model that is intrinsically robust to many types of abnormal node features by design. It improves the performance in various kinds of abnormal scenarios without needing to sacrifice clean accuracy in normal settings.

## 6 Conclusion

In this work, we discover an intrinsic tension between feature aggregation and residual connection in the message passing scheme of GNNs, as well as the corresponding performance tradeoff between nodes with abnormal and normal features. We analyze possible reasons to explain these findings from the perspective of graph Laplacian smoothing. Our understandings further motivate us to propose a simple, efficient, interpretable and adaptive message passing scheme as well as a new GNN model with adaptive residual, named AirGNN. AirGNN provides a node-wise adaptive transition between feature aggregation and residual connection, and the significant advantages of AirGNN are demonstrated through extensive experiments.

## Acknowledgments and Disclosure of Funding

This research is supported by the National Science Foundation (NSF) under grant numbers IIS1714741, CNS1815636, IIS1845081, IIS1907704, DRL2025244, IIS1928278, IIS1955285,

IOS2107215, IOS2035472 and Army Research Office (ARO) under grant number W911NF-21-1-0198.

## Societal Impact and Limitations

The methodology proposed in this paper might have significant positive societal impact since it reduces machine learning models' vulnerability to abnormal datasets that generally exist in real-world applications, especially in many security-critical scenarios. In practice, for a given graph, we do not have the prior knowledge about if the graph is clean, has noisy features or adversarial features. Therefore, algorithms like the proposed AirGNN that can work under both the clean and various abnormal feature settings are appealing. While we are unaware of any potential negative society impact, we point out two limitations of this work: (1) this paper focuses on abnormal node features and has not evaluated the performance of the proposed method when the dataset contains both abnormal node features and edges; (2) it is unclear how it performs on heterophilic graphs. It will be interesting to investigate these problems in future works.

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
