# Appendix for Graph Neural Networks with Adaptive Residual

## A  Convergence Guarantee

In Section 3, we provide the convergence guarantee and practical parameter setting for AMP in Theorem 1. In this section, we first restate the theorem for convenience, followed by the proof.

**Theorem 1** (Convergence of AMP). *Under the stepsize setting $\gamma < \frac{1}{(1-\lambda)\|\tilde{\mathbf{L}}\|_2}$, the proposed adaptive message passing scheme (AMP) in Eq. (9) and Eq. (10) converges to the optimal solution of the problem defined in Eq. (8). In practice, it is sufficient to choose any $\gamma < \frac{1}{2(1-\lambda)}$ since $\|\tilde{\mathbf{L}}\|_2 \leq 2$. Moreover, if the connected components of the graph $\mathcal{G}$ are not bipartite graphs, it is sufficient to choose $\gamma = \frac{1}{2(1-\lambda)}$ since $\|\tilde{\mathbf{L}}\|_2 < 2$.*

*Proof.* The objective that the iterations in AMP try to optimize is

$$\underset{\mathbf{X} \in \mathbb{R}^{n \times d}}{\arg\min} \ \mathcal{L}(\mathbf{X}) := \underbrace{\lambda \|\mathbf{X} - \mathbf{X}_{\text{in}}\|_{21}}_{g(\mathbf{X})} + \underbrace{(1-\lambda)\text{tr}(\mathbf{X}^\top (\mathbf{I} - \tilde{\mathbf{A}})\mathbf{X})}_{f(\mathbf{X})}, \tag{17}$$

where $f$ and $g$ are both convex functions. Moreover, $g$ is a non-smooth function, while $f$ is a smooth function. In particular, $f$ is $L$-smoothness where $L = 2(1-\lambda)\|\tilde{\mathbf{L}}\|_2 = 2(1-\lambda)\|\mathbf{I} - \tilde{\mathbf{A}}\|_2$ due to

$$\|\nabla f(\mathbf{X}_1) - \nabla f(\mathbf{X}_2)\|_F = \|2(1-\lambda)\tilde{\mathbf{L}}(\mathbf{X}_1 - \mathbf{X}_2)\|_F \leq 2(1-\lambda)\|\tilde{\mathbf{L}}\|_2 \|\mathbf{X}_1 - \mathbf{X}_2\|_F. \tag{18}$$

AMP essentially applies a forward-backward splitting on the composite objective $g(\mathbf{X}) + f(\mathbf{X})$:

$$\mathbf{X}^{k+1} = (\mathbf{I} + \gamma \partial g)^{-1}(\mathbf{X}^k - \gamma \nabla f(\mathbf{X}^k)) \tag{19}$$

$$= \underset{\mathbf{X}}{\arg\min} \frac{1}{2}\|\mathbf{X} - (\mathbf{X}^k - \gamma f(\mathbf{X}^k))\|_F^2 + \gamma g(\mathbf{X}), \tag{20}$$

which is known as proximal gradient method. The convergence of this forward-backward splitting is ensured if the stepsize satifies $\gamma < \frac{2}{L}$ according to Lemma 4.4 in [39]. Therefore, AMP provably converges to the optimal solution under the setting $\gamma < \frac{1}{(1-\lambda)\|\tilde{\mathbf{L}}\|_2}$. For the symmetrically normalized Laplacian matrix, we have $\|\tilde{\mathbf{L}}\|_2 \leq 2$ [40] and thus $\frac{1}{2(1-\lambda)} \leq \frac{1}{(1-\lambda)\|\tilde{\mathbf{L}}\|_2}$. Therefore, any $\gamma < \frac{1}{2(1-\lambda)}$ will be sufficient. Moreover, according to [40], if the connected components of the graph $\mathcal{G}$ are not bipartite graphs, we have $\|\tilde{\mathbf{L}}\|_2 < 2$ and thus $\gamma = \frac{1}{2(1-\lambda)} < \frac{1}{(1-\lambda)\|\tilde{\mathbf{L}}\|_2}$ is sufficient.

$\square$

## B  Data Statistics

In the experiments, the data statistics (full graphs) used in Section 4.2 are summarized in Table 3. The data statistics (largest connected components) used in Section 4.3 are summarized in Table 4. We use fixed data splits for Cora, CiteSeer, PubMed and ogbn-arxiv datasets, and random data split for other datasets.

Table 3: Data statistics on benchmark datasets.

| Dataset | Classes | Nodes | Edges | Features | Training Nodes | Validation Nodes | Test Nodes |
|---|---|---|---|---|---|---|---|
| Cora | 7 | 2708 | 5278 | 1433 | 20 per class | 500 | 1000 |
| CiteSeer | 6 | 3327 | 4552 | 3703 | 20 per class | 500 | 1000 |
| PubMed | 3 | 19717 | 44324 | 500 | 20 per class | 500 | 1000 |
| Coauthor CS | 15 | 18333 | 81894 | 6805 | 20 per class | 30 per class | Rest nodes |
| Coauthor Physics | 5 | 34493 | 247962 | 8415 | 20 per class | 30 per class | Rest nodes |
| Amazon Computers | 10 | 13381 | 245778 | 767 | 20 per class | 30 per class | Rest nodes |
| Amazon Photo | 8 | 7487 | 119043 | 745 | 20 per class | 30 per class | Rest nodes |
| obgn-arxiv | 40 | 169343 | 1166243 | 128 | 54% | 18% | 28% |

Table 4: Dataset statistics for adversarially attacked datasets.

| Dataset | $N_{LCC}$ | $E_{LCC}$ | Classes | Features |
|---------|-----------|-----------|---------|----------|
| Cora | 2,485 | 5,069 | 7 | 1,433 |
| CiteSeer | 2,110 | 3,668 | 6 | 3,703 |
| PubMed | 19,717 | 44,338 | 3 | 500 |

# C   Additional Results for the Preliminary Study

In this section, we provide additional results on CiteSeer and PubMed datasets for the preliminary study in Section 2. The results on these two datasets are showed in Figure 8, 9, 10 and 11. It can be observed that residual connection helps obtain better performance on normal features but it is detrimental to abnormal features, which aligns with the findings in Section 2.

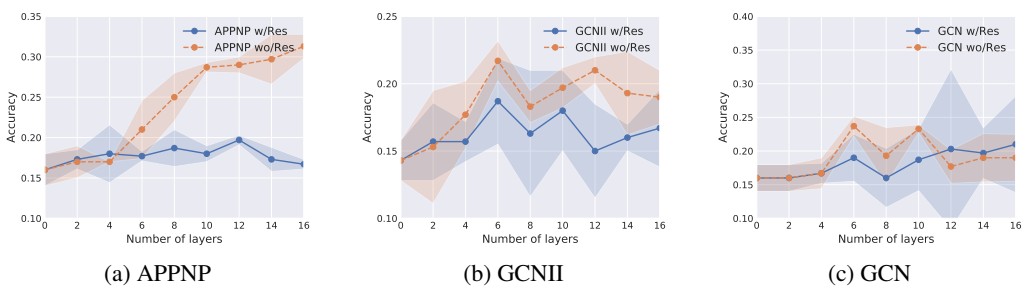

(a) APPNP                    (b) GCNII                    (c) GCN

Figure 8: Node classification accuracy on abnormal nodes (CiteSeer)

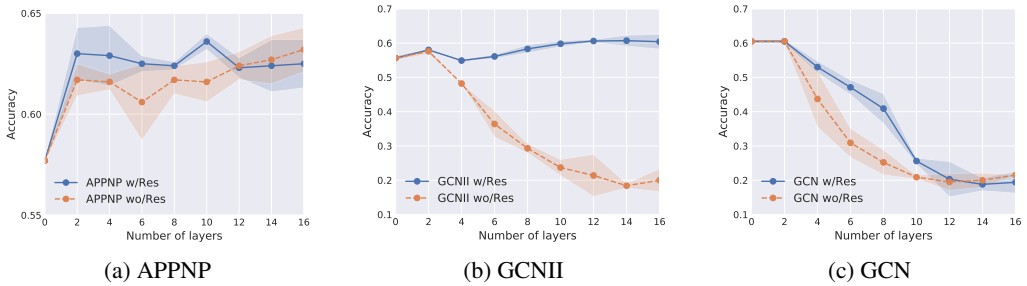

(a) APPNP                    (b) GCNII                    (c) GCN

Figure 9: Node classification accuracy on normal nodes (CiteSeer)

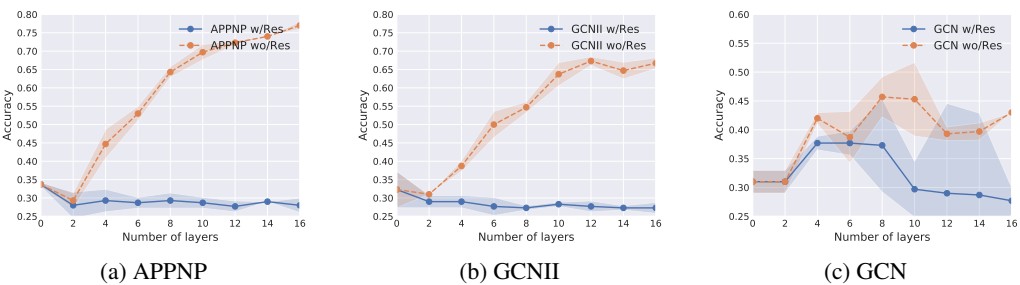

(a) APPNP                    (b) GCNII                    (c) GCN

Figure 10: Node classification accuracy on abnormal nodes (PubMed)

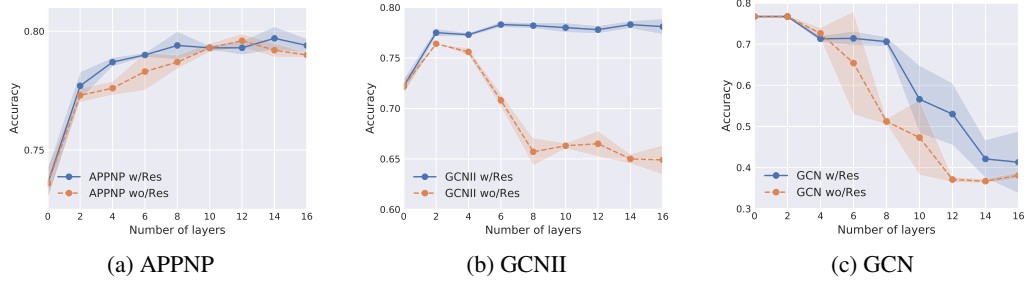

| (a) APPNP | (b) GCNII | (c) GCN |

Figure 11: Node classification accuracy on normal nodes (PubMed)

# D    Additional Experiments for the Proposed Method

In this section, we provide more experiments and ablation study for the proposed AirGNN.

## D.1    Experiments on More Datasets

In this subsection, we provide additional experiments for Section 4. In particular, we conduct the experiments for the noisy feature scenario on the following 5 datasets: Coauthor CS [22], Coauthor Physics [22], Amazon Computers [22], Amazon Photo [22], and ogbn-arxiv [23]. The node classification accuracy are showed in Figures 12, 13, 14, 15, and 16, respectively. Specifically, the accuracy on abnormal nodes and normal nodes are plotted separately in (a) and (b), with respect to the ratio of noisy nodes.

When the ratios of noisy nodes are within a reasonable range, we can observe that (1) AirGNN obtains much better accuracy on abnormal nodes on all datasets, which verifies its stronger resilience to abnormal features; and (2) AirGNN achieves better or sometimes comparable accuracy on normal nodes in most cases, which shows its capability to maintain good performance for normal nodes.

However, when the noise ratio is very high, the performance of AirGNN drops quickly. This is because the modulation hyperparameter $\lambda$ is tuned based on the clean dataset such that it is far away from being optimal for highly noisy dataset. But it can be significantly improved by adjusting the hyperparameter $\lambda$ as discussed in next subsection.

These results suggest the significant advantages of adaptive residual in AirGNN, and confirm the conclusion in the main paper. The adversarial attack on larger graphs is computationally expensive so we omit the results on more datasets in the adversarial feature scenario.

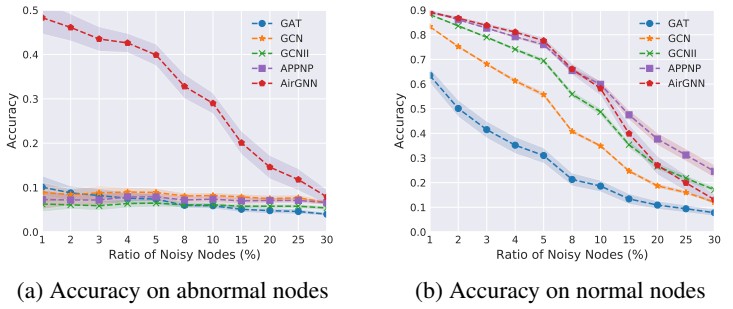

| (a) Accuracy on abnormal nodes | (b) Accuracy on normal nodes |

Figure 12: Node classification accuracy in noisy features scenario (Coauthor CS)

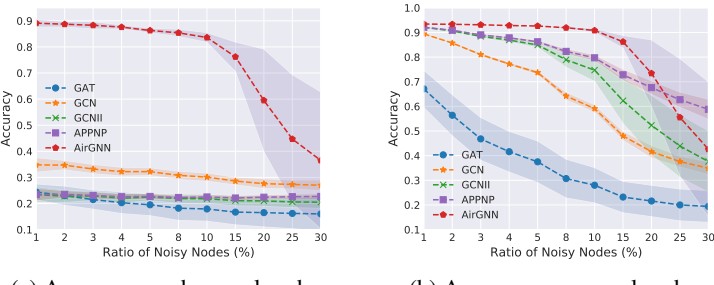

(a) Accuracy on abnormal nodes          (b) Accuracy on normal nodes

Figure 13: Node classification accuracy in noisy features scenario (Coauthor Physics)

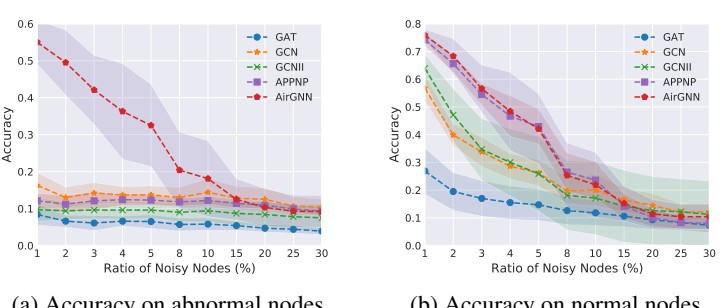

(a) Accuracy on abnormal nodes          (b) Accuracy on normal nodes

Figure 14: Node classification accuracy in noisy features scenario (Amazon Computers)

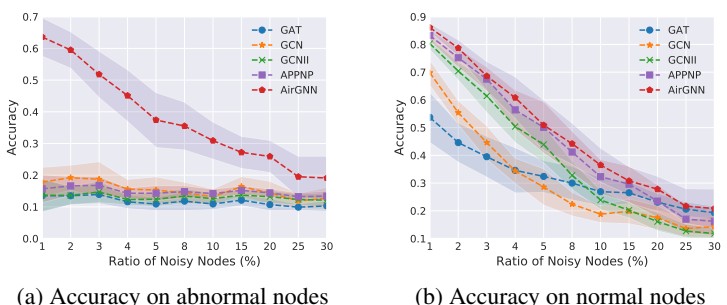

(a) Accuracy on abnormal nodes          (b) Accuracy on normal nodes

Figure 15: Node classification accuracy in noisy features scenario (Amazon Photo)

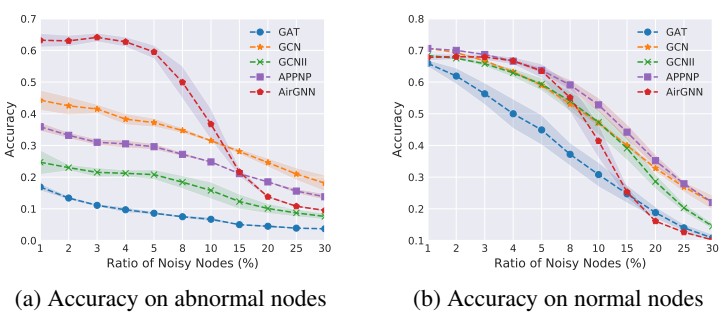

(a) Accuracy on abnormal nodes          (b) Accuracy on normal nodes

Figure 16: Node classification accuracy in noisy features scenario (ogbn-arxiv)

## D.2  AirGNN with Adjusted $\lambda$

Note that in Figures 12, 13, 14,  15, and 16, the performance of AirGNN drops significantly when the noise ratio is very large. This is because the modulation hyperparameter $\lambda$ is tuned based on the clean dataset such that it is far away from being optimal for highly noisy dataset. In fact, the performance of AirGNN can be significantly improved by adjusting $\lambda$ during test time according to the performance on the validation set. Taking the Coauthor CS [22] dataset as an example, we compare AirGNN with APPNP and we tune the hyperparameter $\lambda$ and $\alpha$ for them (denoted as AirGNN-tuned and APPNP-tuned) for a fair comparison as showed in Figure 17. The result verifies that AirGNN-tuned gets tremendous improvement on both abnormal and normal nodes by adjusting $\lambda$. However, APPNP-tuned only focuses on improving global performance and overlooks the abnormal nodes after adjusting $\alpha$ based on validation performance so that the performance on abnormal node are much worse.

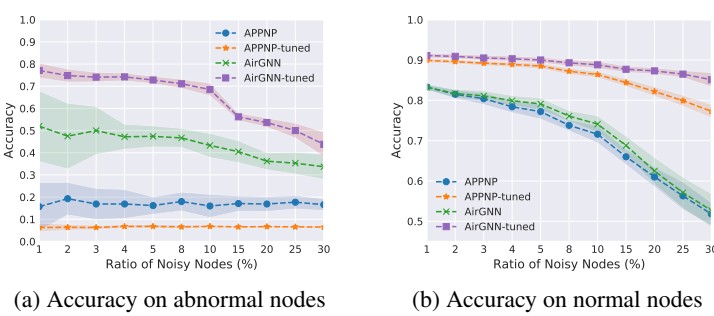

| (a) Accuracy on abnormal nodes | (b) Accuracy on normal nodes |

Figure 17: Node classification accuracy in noisy features scenario with adjustment (Coauthor CS)

## D.3  Detailed Comparison with APPNP

Figure 1 in Section 2 shows that APPNP without residual performs well on the noisy nodes. Therefore, in order to demonstrate the advantages of AirGNN, it is of interest to make a detailed comparison between AirGNN and the two variants of APPNP (w/Res and wo/Res). We evaluate their performance on noisy nodes, normal nodes, and overall nodes on Cora dataset, and the results under varying noise ratio are summarized in Table 5, Table 6, and Table 7. We can make the following observations:

- In Table 5, both AirGNN and APPNP wo/Res significantly outperform APPNP w/Res on noisy nodes, and AirGNN achieves comparable performance with APPNP wo/Res. This verifies that the residual connection in GNN amplifies the vulnerability to abnormal features, and AirGNN is able to adaptively adjust the residual connections for abnormal nodes to reduce the vulnerability.

- In Table 6 and Table 7, AirGNN consistently outperforms APPNP wo/Res, which verifies the importance of residual connections in maintaining good performance on normal nodes. AirGNN exhibits much better performance than APPNP w/Res, which shows the benefits of removing abnormal features by adaptive residual.

- APPNP wo/Res is a special case of AirGNN with $\lambda = 0$. Moreover, as noted in Section D.2, the performance of AirGNN in Table 5, Table 6, and Table 7 can be further improved by adjusting the modulation hyperparameter $\lambda$ for each noise ratio according to validation performance.

As discussed in Section 3, in existing GNNs such as APPNP and GCNII, the conflict between feature aggregation and residual connection can only be partially mitigated by adjusting the residual weight $\alpha$. However, such global adjustment cannot be adaptive to a subset of the nodes, which explains the advantages of AirGNN in above observations. In the adversarial feature setting, we can make similar observations but here we omit the comparison.

Table 5: Comparison between APPNP and AirGNN on abnormal (noisy) nodes (Cora).

| Noisy ratio | 5% | 10% | 15% | 20% | 25% | 30% |
|---|---|---|---|---|---|---|
| APPNP w/Res | $0.167 \pm 0.034$ | $0.170 \pm 0.070$ | $0.170 \pm 0.027$ | $0.193 \pm 0.031$ | $0.187 \pm 0.024$ | $0.178 \pm 0.026$ |
| APPNP wo/Res | $0.469 \pm 0.035$ | $0.442 \pm 0.062$ | $0.427 \pm 0.038$ | $0.381 \pm 0.043$ | $0.383 \pm 0.045$ | $0.354 \pm 0.067$ |
| AirGNN | $0.474 \pm 0.048$ | $0.433 \pm 0.055$ | $0.405 \pm 0.050$ | $0.362 \pm 0.039$ | $0.353 \pm 0.050$ | $0.337 \pm 0.057$ |

Table 6: Comparison between APPNP and AirGNN on normal nodes (Cora).

| Noisy ratio | 5% | 10% | 15% | 20% | 25% | 30% |
|---|---|---|---|---|---|---|
| APPNP w/Res | $0.773 \pm 0.015$ | $0.712 \pm 0.024$ | $0.669 \pm 0.019$ | $0.622 \pm 0.024$ | $0.580 \pm 0.032$ | $0.530 \pm 0.029$ |
| APPNP wo/Res | $0.761 \pm 0.014$ | $0.709 \pm 0.025$ | $0.664 \pm 0.015$ | $0.599 \pm 0.025$ | $0.556 \pm 0.035$ | $0.497 \pm 0.049$ |
| AirGNN | $0.791 \pm 0.015$ | $0.741 \pm 0.021$ | $0.688 \pm 0.024$ | $0.625 \pm 0.034$ | $0.571 \pm 0.039$ | $0.527 \pm 0.042$ |

Table 7: Comparison between APPNP and AirGNN on all nodes (Cora).

| Noisy ratio | 5% | 10% | 15% | 20% | 25% | 30% |
|---|---|---|---|---|---|---|
| APPNP w/Res | $0.743 \pm 0.015$ | $0.657 \pm 0.026$ | $0.594 \pm 0.017$ | $0.536 \pm 0.024$ | $0.482 \pm 0.025$ | $0.425 \pm 0.025$ |
| APPNP wo/Res | $0.746 \pm 0.013$ | $0.682 \pm 0.026$ | $0.628 \pm 0.015$ | $0.556 \pm 0.027$ | $0.513 \pm 0.034$ | $0.455 \pm 0.053$ |
| AirGNN | $0.775 \pm 0.015$ | $0.710 \pm 0.021$ | $0.646 \pm 0.025$ | $0.572 \pm 0.033$ | $0.516 \pm 0.038$ | $0.470 \pm 0.044$ |

## D.4 Comparison with Robust Model

To further demonstrate the advantages of the proposed AirGNN, we compare it with a representative robust model, Robust GCN [24]. Tables 18, 19 and 20 show the performance comparison between Robust GCN and AirGNN on Cora, Citeseer and PubMed, respectively. The accuracy on abnormal nodes and normal nodes are plotted separately in (a) and (b), with respect to the ratio of noisy nodes. These figures show that AirGNN achieves significant better performance than Robust GCN on both abnormal and normal nodes in the noisy feature scenario.

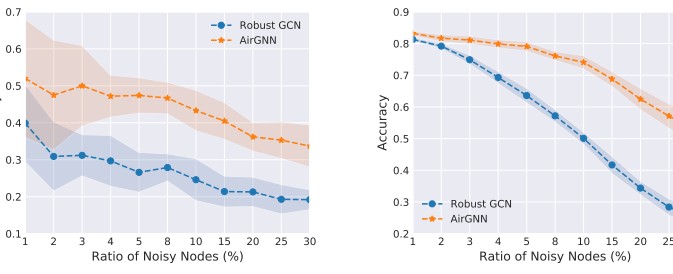

(a) Accuracy on abnormal nodes         (b) Accuracy on normal nodes

Figure 18: Node classification accuracy in noisy features scenario (Cora)

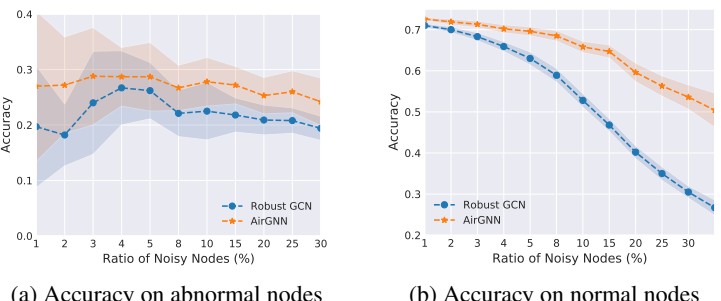

(a) Accuracy on abnormal nodes         (b) Accuracy on normal nodes

Figure 19: Node classification accuracy in noisy features scenario (CiteSeer)

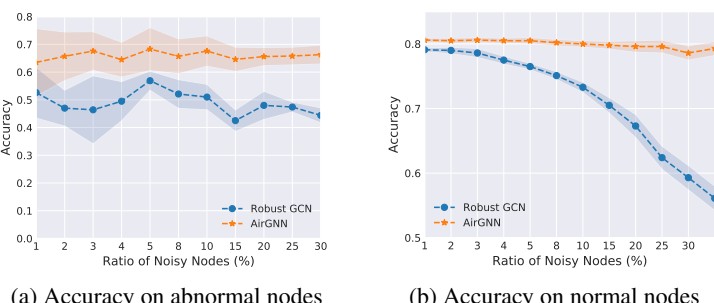

(a) Accuracy on abnormal nodes

(b) Accuracy on normal nodes

Figure 20: Node classification accuracy in noisy features scenario (PubMed)

## D.5 Performance in the Clean Setting

Table 8 shows the overall performance when the dataset does not contain abnormal node features. The performance of APPNP and AirGNN are comparable, which supports that AirGNN doesn't need to sacrifice clean performance for better robustness. AirGNN also outperforms Robust GCN in the clean data setting.

Table 8: Comparison between AirGNN, APPNP, and Robust GCN in the clean setting.

| Dataset | Cora | CiteSeer | PubMed |
|---|---|---|---|
| Robust GCN | $0.817 \pm 0.005$ | $0.710 \pm 0.005$ | $0.791 \pm 0.003$ |
| APPNP | $0.842 \pm 0.004$ | $0.719 \pm 0.004$ | $0.804 \pm 0.003$ |
| AirGNN | $0.839 \pm 0.004$ | $0.726 \pm 0.004$ | $0.806 \pm 0.003$ |