# OpenReview forum: "Graph Neural Networks with Adaptive Residual"
_NeurIPS.cc/2021/Conference — NeurIPS 2021 Poster_

### Official Review · Reviewer_AKM9 · 2021-07-08

**Rating:** 6
**Confidence:** 4

**Summary:**

This paper examines the behavior of GNNs when node features are perturbed during test time.  AirGNN is proposed based on the observations from the standard GNNs. AirGNN adaptively computes the weight of the initial embedding and aggregated embeddings and is robust to random/adversarial noise keeping the standard classification performance.

**Limitations And Societal Impact:**

No limitations are discussed in the manuscript. It is important to discuss the cons against APPNP as the proposed method is a variant of it. What information is lost using the L21 norm instead of L2? As the proposed method is related to adversarial attacks, the societal impact can also be discussed. Is the proposed method reliable enough to deploy in the real world?

**Main Review:**

# Strengthes

* The proposed method is well motivated (Sections 2.1, 2.2, and 3.1). These sections argue that deep GNNs without residual connections are too smooth, whereas GNNs with residual connections can rely on noisy features. The proposed method tries to select the better scenario adaptively. This is reasonable.
* Section 3.3 provides interpretations of the proposed method. These explanations are intuitive and helpful for understanding.

# Weaknesses

* Many robust GNNs have been proposed [A, B]. These methods are not discussed or compared with the proposed method. I would like to see qualitative discussions and quantitative comparisons with them.
* The proposed method is a variant of APPNP. The proposed method adopts the L21 norm instead of the Frobenius norm. It is important to make more exhaustive comparisons with APPNP. See also the questions below.

# Questions

* Figure 1 (a) says APPNP wo/Res achieves accuracy 0.6 on abnormal features, but Figure 5 (a) says APPNP's accuracy is around 0.2. Why does this happen? Is APPNP w/Res used in Figure 5 (a)? (as a result of hyperparameter tuning?) I would like to see the performance of APPNP wo/Res in all settings. As far as I understand, AirGNN should perform worse on abnormal features than APPNP w/Res while it should perform better on normal features. This should be clarified in the experiments.
* See also Limitations And Societal Impact below.

## Reference

* [A] Robust Graph Convolutional Networks Against Adversarial Attacks. KDD 2019.
* [B] Adversarial attack and defense on graph data: A survey. https://arxiv.org/abs/1812.10528

**Time Spent Reviewing:**

4

---

> ### Author Response · Authors · 2021-08-10
> **Response to all concerns by Reviewer AMK9**
>
> Dear reviewer,
>
> Thanks for the positive and valuable feedback.  Here we provide detailed responses to address the concerns in your comments, and we will carefully incorporate them in the revision.
>
> **Question 1: Many robust GNNs are not compared with the proposed method. Would like to see qualitative discussion and quantitative comparisons with them.**
>
> Answer: (1) In practice, for a given graph, we do not have the prior knowledge about if the graph is clean, has noisy features or has adversarial features. Therefore, algorithms like the proposed AirGNN that can work under multiple abnormal feature settings are desired. However, these GNN defense models have been specifically designed for the adversarial setting. Thus choosing them as the baselines and evaluating them under multiple settings seem unfair to them. That is the major reason we exclude them in the current version.
>
> (2) As suggested by the reviewer, including these defense methods and showing the performance comparison can further demonstrate the advantages of the proposed AirGNN. Thus, we follow the suggestion and conduct experiments to include several representative robust models, including Robust GCN, GCN-Jaccard and GCN-SVD. The comparison in the noise and adversarial settings on Cora are shown in the following tables:
>
> | Ratio of noisy nodes | 1% | 2% | 3% |  4% |  5% |  8% |  10% |  15% | 20% |  25% | 30% |
> | - | - | - | - | - | - | - | - | - | - | - | - |
> | **Abnormal nodes** |
> | RGCN | 0.497 | 0.473 | 0.479 | 0.360 | 0.268 | 0.335 | 0.236 | 0.272 | 0.204 | 0.215 | 0.201 |
> | GCN-SVD | 0.200 | 0.200 | 0.150 | 0.238 | 0.160 | 0.250 | 0.170 | 0.223 | 0.157 | 0.144 | 0.150 |
> | GCN-Jaccard | 0.040 | 0.220 | 0.193 | 0.118 | 0.136 | 0.189 | 0.143 | 0.152 | 0.150 | 0.137 | 0.127 |
> | AirGNN | 0.650 | 0.511 | 0.485 | 0.502 | 0.496 | 0.421 | 0.463 | 0.392 | 0.363 | 0.396 | 0.343 |
> | **Normal nodes** |
> | RGCN | 0.800 | 0.767 | 0.727 | 0.679 | 0.636 | 0.569 | 0.500 | 0.410 | 0.330 | 0.290 | 0.251 |
> | GCN-SVD | 0.674 | 0.621 | 0.537 | 0.458 | 0.387 | 0.322 | 0.292 | 0.267 | 0.253 | 0.217 | 0.197
> | GCN-Jaccard | 0.762 | 0.742 | 0.716 | 0.695 | 0.664 | 0.617 | 0.560 | 0.466 | 0.399 | 0.339 | 0.261 |
> | AirGNN | 0.834 | 0.809 | 0.804 | 0.804 | 0.791 | 0.774 | 0.721 | 0.690 | 0.626 | 0.601 | 0.565 |
>
> | Perturbation budget | 1  |  2  |  5  |  10  |  20  |  50  |  80 |
> | - | - | - | - | - | - | - | - |
> | RGCN | 0.800 | 0.775 | 0.717 | 0.633 | 0.558 | 0.477 | 0.458 |
> | GCN-SVD | 0.742 | 0.717 | 0.642 | 0.567 | 0.442 | 0.383 | 0.325 |
> | GCN-Jaccard | 0.775 | 0.717 | 0.633 | 0.517 | 0.317 | 0.133 | 0.125 |
> | AirGNN | 0.850 | 0.832 | 0.812 | 0.800 | 0.738 | 0.680 | 0.650 |
>
> The results from the tables highlight one big advantage of AirGNN: AirGNN does not assume the type of abnormal features, and it can work significantly better than these robust models under different types of abnormal features. These results clearly justify the advantages of AirGNN. We will extensively include more related works and results in the revision.
>
> **Question 2: Figure 1 (a) says APPNP wo/Res achieves accuracy 0.6 on abnormal features, but Figure 5 (a) says APPNP's accuracy is around 0.2. Why does this happen? Is APPNP w/Res used in Figure 5 (a) as a result of hyperparameter tuning?**
>
> **Answer**: (1) First, we would like to clarify that the results in Figure 1 and Figure 5 are not directly comparable since the nodes being selected to be perturbed are different. Second, we agree that APPNP wo/Res performs better than APPNP w/Res on abnormal features. But the performance on normal features will be much worse, as observed and discussed in the preliminary study (i.e., Figure 1(a) and Figure 2(b)).
>
> (2) In practice we typically do not know which node is abnormal or normal. Thus, in the experimental section, we tune APPNP based on the overall performance of all nodes, and APPNP wo/Res is not selected by hyperparameter tuning although it is included in the search space. In other words, as shown in (1), APPNP wo/Re can work better for abnormal features but will significantly hurt normal features; thus, overall, APPNP wo/Re is not optimal. This observation suggests we indeed need an algorithm that can naturally handle the tension between aggregation and residual. That is the major motivation to design AirGNN.
>
> (3) As mentioned in (2), the motivation of AirGNN is exactly to reduce the tension between feature aggregation and residual connection naturally and automatically (for instance between APPNP w/Res and APPNP wo/Res). See the motivation between line 142-144 in Section 3.1:
>
> > Although this conflict can be partially mitigated by adjusting the residual connection such as the residual weight $\alpha$ in GCNII and APPNP, such global adjustment cannot be adaptive to a subset of the nodes, e.g., the nodes with abnormal features.
>
> This is to say, APPNP wo/Res might improve the performance on abnormal nodes, but the price is to sacrifice the performance on normal nodes as well as the global performance. (This will be verified in the answer to Question 3)
>
> By the proposed adaptive message passing (AMP), AirGNN is able to significantly improve the performance on abnormal nodes while maintaining good performance on normal nodes. The adaptive residual weights evaluated in Section 4.4 explain how AirGNN works. Table 1 and Table 2 in Section 4.4 show that the average residual weight for abnormal nodes is very close to 0, which reduces the impact of abnormal features. While the average residual weight for normal nodes is larger, which avoids over smoothing and maintains good overall performance. Such node-wise adaptivity is the key advantage of AMP and AirGNN.
>
> **Question 3: AirGNN should perform worse on abnormal features than APPNP w/Res while it should perform better on normal features. This should be clarified in the experiments. It is important to make more exhaustive comparisons with APPNP, such as APPNP wo/Res.**
>
> **Answer**:
> Following the logic in Question 2, we believe there is a typo in the comment: APPNP w/Res should be APPNP wo/Res. The intended comment should be
>
> >AirGNN should perform worse on abnormal features than APPNP wo/Res while it should perform better on normal features.
>
> Please let us know if we misunderstand this. But if this is the comment, we have the following response:
>
> (1) Following the response to Question 2, we include a comparison between AirGNN, APPNP w/Res and APPNP wo/Res on Cora to demonstrate the advantages of AirGNN in the table:
>
> | **Ratio of noisy nodes** | 1% | 2% | 3% |  4% |  5% |  8% |  10% |  15% | 20% |  25% | 30% |
> | :-: | :-: | :-: | :-: | :-: | :-: | :-: | :-: | :-: | :-: | :-: | :-: |
> | **Abnormal nodes** |
> | APPNP w/Res | 0.110 | 0.145 | 0.133 | 0.195 | 0.138 | 0.148 | 0.156 | 0.139 | 0.145 | 0.186 | 0.183 |
> | APPNP wo/Res | 0.640 | 0.550 | 0.490 | 0.368 | 0.426 | 0.461 | 0.372 | 0.385 | 0.325 | 0.353 | 0.301 |
> | AirGNN | 0.650 | 0.485 | 0.470 | 0.502 | 0.496 | 0.421 | 0.463 | 0.392 | 0.363 | 0.396 | 0.343 |
> | **Normal nodes** |
> | APPNP w/Res | 0.836 | 0.809 | 0.796 | 0.785 | 0.771 | 0.748 | 0.687 | 0.644 | 0.599 | 0.580 | 0.553 |
> | APPNP wo/Res | 0.808 | 0.778 | 0.762 | 0.757 | 0.747 | 0.710 | 0.695 | 0.605 | 0.503 |0.483 | 0.441 |
> | AirGNN | 0.834 | 0.809 | 0.804 | 0.804 | 0.791 | 0.774 | 0.721 | 0.690 | 0.626 | 0.601 | 0.565 |
> | **Overall performance** |
> | APPNP w/Res | 0.829 | 0.796 | 0.776 | 0.761 | 0.739 | 0.700 | 0.634 | 0.568 | 0.508 | 0.481 | 0.442 |
> | APPNP wo/Res | 0.806 | 0.773 | 0.754 | 0.742 | 0.731 | 0.690 | 0.663 | 0.572 | 0.467 | 0.450 | 0.399 |
> | AirGNN | 0.833 | 0.802 | 0.794 | 0.792 | 0.776 | 0.746 | 0.696 | 0.646 | 0.573 | 0.550 | 0.499 |
>
> We can observe that AirGNN significantly outperforms APPNP wo/Res on normal features. However, AirGNN is not necessarily worse than APPNP wo/Res on abnormal features, and it sometimes even outperforms APPNP wo/Res. This is because AirGNN is able to reduce the impact of abnormal features on neighboring nodes, while APPNP wo/Res will fully propagate the abnormal features to neighbors.
>
> (2) We also highlight that APPNP wo/Res is also a special case of AirGNN if we set $\lambda=0$ in AirGNN. Therefore, AirGNN can at least achieve the same performance as APPNP wo/Res, if we set $\lambda=0$. However, this setting is not very useful because the overall performance will be much worse, as showed in the above table.
>
> **Question 4: The limitations and society impact are not discussed. What information is lost using the L21 norm instead of L2?  Is the proposed method reliable enough to deploy in the real world?**
>
> **Answer**: (1) Using L21 norm instead of L2 norm will increase the resilience to abnormal features. However, one potential limitation is that this change might impact the performance on clean datasets (i.e., there is no abnormal feature). We include such comparison on clean datasets as showed in the table:
>
> | Dataset | Cora | CiteSeer | PubMed | Photo | Physics |
> | :-: | :-: | :-: | :-: | :-: | :-: |
> | APPNP | 0.842 | 0.719 | 0.804 | 0.901 | 0.936 |
> | AirGNN | 0.839 | 0.726 | 0.806 | 0.906 | 0.936 |
>
> Surprisingly, using L21 norm in AirGNN achieves comparable performance with APPNP which uses L2 norm. Therefore, it suggests that using the L21 norm might not lose information for achieving good performance.
>
> (2) The proposed method is reliable to deploy in real-world because AirGNN is intrinsically robustness to abnormal features. It improves the performance on various kinds of abnormal features. Importantly, it maintains good performance on normal features. As a comparison, many robust GNN models will sacrifice the performance when the datasets are clean. Therefore, the adaptive message passing can be used as a general message passing scheme for all GNN models to improvde the robustness without hurting the overall performance.
>
> Overall, we hope that we have addressed the concerns in your comments, and please kindly let us know if there is any further concern, and we are happy to clarify.

---

> > ### Comment · Reviewer_AKM9 · 2021-08-30
> > **Thank you for the detailed response**
> >
> > Thank you for the detailed response.
> >
> > I'm still not sure why the performance of APNNP wo/Res reported here is so poor. Figure 1 (a) says, with 10% noise, APNNP wo/Res achieves > 0.6 accuracy for abnormal nodes and > 0.7 accuracy for normal nodes. However, the table in the response for Question 3 says APNNP wo/Res achieves 0.372 accuracy for abnormal nodes with 10% noise. The authors stated that `we would like to clarify that the results in Figure 1 and Figure 5 are not directly comparable since the nodes being selected to be perturbed are different.`, but Section 4.2 says that the noise is generated similarly to the preliminary study. So I assume that the difference in the node selection came from stochasticity. I feel the difference of performances is too large if the difference is solely caused by the selection of noise nodes. The difference is out of the confidence interval reported in Figure (a). This inconsistent result is harming the credibility of experimental results. It would be variable to inspect what caused such a large difference in Figure 1 (a) and this table more carefully.
> >
> > Due to the remaining concern, I keep my initial score. I believe more careful and thorough discussions on APPNP will improve the value of this paper.

---

> > > ### Author Response · Authors · 2021-08-31
> > > **Further clarification**
> > >
> > > Dear reviewer,
> > >
> > > Thanks for your feedback. Your observations and  suggestions are very helpful and valuable, which helps us improve the paper. Next we will clarify your concerns in the feedback.
> > >
> > > The performance difference between Figure 1(a) and what is reported in our response for Question 3 is indeed caused by the selection of noise nodes. To verify this, we repeat random node selections (10\%) 10 times, and the performance on abnormal nodes for APPNP wo/Res is 0.46 ± 0.06 (mean ± standard deviation). From the standard deviation (0.06), we can tell the performance 0.58 in Figure 1(a) and 0.37 in our response are in a reasonable range.
> > >
> > > Intuitively, such large deviation is also reasonable since some nodes are less vulnerable to abnormal features while others are more vulnerable. For instance, it is easy to correct the abnormal features by feature propagation for high-degree nodes since the impact of self-loop is small, while it is difficult to correct the abnormal features for low-degree nodes since the impact of self-loop is large. With 10\% selection, the selected node sets can have very diverse characteristics.
> > >
> > > We also note that the confidence intervals showed in the figures only reflect the randomness of different initialization of GNN models but they do not reflect the randomness of node selection since only one random selection is chosen. In fact, AirGNN consistently outperforms others in numerous random selections so we only report the performance on one fixed selection to ease the replication of our experimental results. However, we do ensure that in the experiment section, the selection is the same for all algorithms so that the comparison is fair. Therefore, the advantages of AirGNN showed in the paper and in our responses are properly evaluated.
> > >
> > > We hope this can address your concern and we will include all these clarifications in the revision.
> > >
> > > Thank you,
> > >
> > > Authors

---

> > > > ### Comment · Reviewer_AKM9 · 2021-08-31
> > > > **Thank you for the response**
> > > >
> > > > Thank you for the response.
> > > >
> > > > Now I know the large difference in performances came from the randomness of node selections. However, if so, I feel the experimental design can be improved.
> > > >
> > > > Two-sigma deviations may not be rare, but the directions of these deviations are advantageous for the authors' claims, i.e., the performance of Figure 1 (a) deviates upward to support Finding 1 (L. 95), and the performance of APPNP w/o Res in the table deviates downward to support that AirGNN outperforms APPNP. These deviations are as large as two sigmas. I cannot wipe out my suspicion that these results involve (intentional or unintentional) cherry-picking. Of course, it is impossible to prove or disprove this suspicion. This paper should have adopted more robust experimental designs so that such a suspicion would not rise.
> > > >
> > > > For example, this paper should have reported the means and confidence intervals with respect to node selections and clarified it, provided the node selections affect the performance so much. The authors claimed `we only report the performance on one fixed selection to ease the replication of our experimental results.`  IMHO, however, clarifying large variances of node selections eases the replication (e.g., in debugging) and is more beneficial for the following researchers rather than reporting only one fixed selection.
> > > >
> > > > In the current form, it is difficult to see how node selections affect the performances and variances in various experiments in the paper and discussions. I recommend including more detailed analyses on the effect of node selections in the next submission or camera ready.

---

> > > > > ### Author Response · Authors · 2021-08-31
> > > > > **Further clarification 2**
> > > > >
> > > > >
> > > > > Dear reviewer,
> > > > >
> > > > > Thanks for your quick reply and we really appreciate that you carefully read our responses.
> > > > >
> > > > > We can understand your suspicion regarding the directions of the two deviations. We want to clarify that (1) the fixed selection is randomly selected among numerous selections; (2) we have consistent observations among different selections; and (3) in our evaluation, randomness can be from both the node selection and the model initialization; and since we have consistent observations among different selections, in our current results, we show the results on one fixed selection to understand the randomness from the model initialization.
> > > > >
> > > > > Also, we want to point out that even with much lower performance (such as 0.37) in Figure 1 (a), it already supports Finding 1. We agree with you that including the results about the impact of node selection will exclude such suspicion. Below we quickly show: the performance improvement of AirGNN over APPNP wo/Res is very significant even on the average over 10 times random node selections (10%):
> > > > >
> > > > > APPNP wo/Res: 0.69 ± 0.02 (overall), 0.46 ± 0.06 (abnormal nodes), 0.71 ±0.02 (normal nodes)
> > > > >
> > > > > AirGNN: 0.73 ± 0.02 (overall), 0.47 ± 0.03 (abnormal nodes), 0.76 ± 0.02 (normal nodes)
> > > > >
> > > > > In the above result, $\lambda$ in AirGNN is tuned from the setting of 1% noise ratio. If we tune $\lambda$ by validation accuracy on this specific noise ratio 10%, we have:
> > > > >
> > > > > AirGNN: 0.76 ± 0.02 (overall), 0.49 ± 0.03 (abnormal nodes), 0.79 ± 0.01 (normal nodes).
> > > > >
> > > > > We will include detailed analyses on the effect of node selection as you suggest in the revision. Moreover, we will make our implementation publically available for reproducibility.
> > > > >
> > > > > Finally, we really enjoy the meaningful discussion with you. Thank you for your help and support!
> > > > >
> > > > > Authors

---

### Official Review · Reviewer_tp11 · 2021-07-11

**Rating:** 6
**Confidence:** 3

**Summary:**

This paper discusses how message passing and feature aggregation are affected by abnormal node features, and designs a message passing module called Adaptive Message Passing (AMP) with adaptive residual connection and feature aggregation. In this way, the module can adaptively deal with the node that is inconsistent with the local feature during the message passing. The paper proposes the AirGNN that is built up with AMP as the message-passing layer in GNN, the experiment result shows that the AirGNN gets a more robust result on both the abnormal node and normal node classification comparing with the baselines.

**Limitations And Societal Impact:**

The authors do not give a section that discusses the limitations and potential negative societal impact. However, since the author stresses that the abnormal node feature exists a lot in the real-world dataset, I may expect the author has a brief discussion on this aspect.

**Main Review:**

The paper gives a solution for dealing with the abnormal node feature in the graph. The authors first give an observation of how the residual connection and number of layers affect the models' performance when the layer grows. Then the author gives the explanation and their solutions to balance the feature aggregation and residual connections affect during the message passing. The experiment result shows their advantages on various datasets comparing with the baselines.

The paper writes in good order, the discussion at the beginning of the paper helps a lot for me to understand the motivation of the paper. The authors give a straightforward solution by designing an adaptive module to control the feature aggregation and residual connection to deal with the abnormal feature in the graph. I am not sure if it is a novel method but seems reasonable to me.

For the experiment part, the authors compared their method with GCN, GAT, APPNP, and GCNII and evaluate the performance on both normal and abnormal features. From the plot, the AirGNN consistently outperforms other baselines when an abnormal feature exists and increases. Overall, the experiment shows the model's advantage properly.

My questions and concerns are the follows:

1. Authors use a single noise setting in the whole paper, I think for this setting, the noise part is an important thing to understand, however, authors do not even write down the detailed noise setting ( what is the mean and var of the noise distribution). I am expecting to see a careful ablation study towards the noise to see if the method is robust to detect variations of the noise feature.

2. I see from the appendix that for other datasets like Coauthor Physics, Amazon photo, obg-arxiv, the AirGNN is pretty close or even worse than other methods like APPNP on the normal features. Do you have any explanation about this? I not confident for the statement that the model can deal with the real-world Graph with abnormal features. Combining with the first suggestions, it is necessary to see if the proposed model performs well when the noise setting changes.

3. I am not an expert in this area, but it seems that from the related works and the baseline, the authors do not mention some competing ''Robust GNN'' baselines.  I believe there should have some adversarial GNN paper that studies the noise feature during the training process.


**Time Spent Reviewing:**

4

---

> ### Author Response · Authors · 2021-08-10
> **Response to all concerns by Reviewer tp11**
>
> Dear reviewer,
>
> Thanks for the recognition of our observations, the writing, and the advantages of the proposed method. Next we provide detailed responses to address your concerns.
>
> **Question 1: The noise setting is not clear. What is the mean and var of the noise distribution? Expect to see a careful ablation study towards the noise to see if the method is robust to detect variations of the noise feature. It is necessary to see if the model performs well when the noise setting changes.**
>
> **Answer**: (1) In section 4.2 (line 248), it is stated that the noisy features are sampled from a multivariate standard Gaussian distribution, which means the mean is 0 and variance is 1.
>
> (2) Following your suggestions, we include two more noise settings for a careful ablation study. The first noise setting is zero-mean Gaussian distribution under different standard deviations, and the results are showed in the following table:
>
> | Std of noise | 0.1 | 0.5 | 1.0 | 1.5 |
> | - | - | - | - | - |
> | **Abnormal nodes** |
> |GCN| 0.414 | 0.204 | 0.125 | 0.167 |
> |GAT| 0.348 | 0.123 | 0.114 | 0.117 |
> |APPNP| 0.442 | 0.203 | 0.156 | 0.156 |
> |AirGNN| 0.646 | 0.525 | 0.463 | 0.328 |
> | **Normal nodes** |
> |GCN| 0.788 | 0.721 | 0.615 | 0.628 |
> |GAT| 0.798 | 0.593 | 0.490 | 0.508 |
> |APPNP| 0.831 | 0.777 | 0.687 | 0.674 |
> |AirGNN| 0.831 | 0.795 | 0.721 | 0.694 |
>
> The second noise setting is noisy under multiple Gaussian distributions. Specifically, we synthesize random noise from three Gaussian distributions with different standard deviations (1,2, and 3), and randomly add these noises to the selected node. The performance is showed in the following table:
>
> | Ratio of noisy nodes | 1% | 2% | 3% |  4% |  5% |  8% |  10% |  15% | 20% |  25% | 30% |
> | :-: | :-: | :-: | :-: | :-: | :-: | :-: | :-: | :-: | :-: | :-: | :-: |
> | **Abnormal nodes** |
> | GCN | 0.140 | 0.170 | 0.100 | 0.120 | 0.138 | 0.146 | 0.104 | 0.162 | 0.110 | 0.170 | 0.143 |
> | GAT | 0.04 | 0.170 | 0.133 | 0.102 | 0.156 | 0.184 | 0.115 | 0.127 | 0.104 | 0.103 | 0.134 |
> | APPNP | 0.07 | 0.215 | 0.090 | 0.110 | 0.226 | 0.224 | 0.133 | 0.169 | 0.156 | 0.168 | 0.174 |
> | AirGNN | 0.640 | 0.455 | 0.427 | 0.485 | 0.364 | 0.419 | 0.292 | 0.305 | 0.315 | 0.272 | 0.258 |
> | **Normal nodes** |
> | GCN | 0.763 | 0.767 | 0.720 | 0.709 | 0.661 | 0.621 | 0.589 | 0.514 | 0.452 | 0.439 | 0.348 |
> | GAT | 0.714 | 0.720 | 0.670 | 0.664 | 0.602 | 0.529 | 0.518 | 0.412 | 0.371 | 0.337 | 0.336 |
> | APPNP | 0.811 | 0.776 | 0.750 | 0.733 | 0.759 | 0.689 | 0.608 | 0.527 | 0.486 | 0.453 | 0.440 |
> | AirGNN | 0.839 | 0.782 | 0.805 | 0.794 | 0.782 | 0.740 | 0.724 | 0.608 | 0.605 | 0.517 | 0.438 |
>
> From these results, we can clearly observe that AirGNN significantly outperform others consistently, which verifies that AirGNN is robust to detect the variations of the noise features.
>
> **Question 2: In the appendix, for datasets like Physics, photo, obgn-arxiv, the performance on normal features of AirGNN is pretty close or even worse than other methods like APPNP. Do you have any explanation about this?**
>
> **Answer**: Thanks for pointing this out. This can be explained as the result of hyperparameter tuning. In order to save computation cost, we tune the modulation hyperparameter $\lambda$ for AirGNN based the performance on a fix noise ratio (1\%) and apply this setting (as summarized in Table 5 in the appendix) to all noise ratios. However, when the noise ratio becomes larger, this setting is not optimal for AirGNN anymore, which explains the decreasing performance. If we tune this modulation hyperparameter for each noise ratio, we can obtain the following comparison on Photo and Physics respectively:
>
>
> | Ratio of noisy nodes | 1% | 2% | 3% |  4% |  5% |  8% |  10% |  15% | 20% |  25% | 30% |
> | :-: | :-: | :-: | :-: | :-: | :-: | :-: | :-: | :-: | :-: | :-: | :-: |
> | **Abnormal nodes** |
> | APPNP | 0.106 | 0.132 | 0.107 | 0.147 | 0.156 | 0.128 | 0.128 | 0.126 | 0.133 | 0.117 | 0.128 |
> | AirGNN | 0.854 | 0.830 | 0.721 | 0.745 | 0.724 | 0.687 | 0.621 | 0.399 | 0.403 | 0.378 | 0.328 |
> | **Normal nodes** |
> | APPNP | 0.820 | 0.817 | 0.798 | 0.800 | 0.795 | 0.764 | 0.750 | 0.705 | 0.688 | 0.647 | 0.612 |
> | AirGNN | 0.906 | 0.904 | 0.896 | 0.893 | 0.892 | 0.877 | 0.871 | 0.835 | 0.831 | 0.832 | 0.779 |
>
> | Ratio of noisy nodes | 1% | 2% | 3% |  4% |  5% |  8% |  10% |  15% | 20% |  25% | 30% |
> | :-: | :-: | :-: | :-: | :-: | :-: | :-: | :-: | :-: | :-: | :-: | :-: |
> | **Abnormal nodes** |
> | APPNP | 0.302 | 0.292 | 0.300 | 0.291 | 0.206 | 0.203 | 0.201 | 0.208 | 0.205 | 0.200 | 0.203 |
> | AirGNN | 0.884 | 0.878 | 0.871 | 0.888 | 0.870 | 0.856 | 0.797 | 0.832 | 0.799 | 0.748 | 0.712 |
> | **Normal nodes** |
> | APPNP | 0.932 | 0.920 | 0.905 | 0.892 | 0.898 | 0.895 | 0.893 | 0.888 | 0.878 | 0.873 | 0.868 |
> | AirGNN | 0.935 | 0.932 | 0.929 | 0.929 | 0.924 | 0.917 | 0.889 | 0.907 | 0.891 | 0.887 | 0.880 |
>
> Note that in the above tables, the hyperparameter of APPNP is well-tuned for each noise ratio as well, so it is a fair comparison. We can observe that AirGNN always outperforms APPNP in all noise ratios, and the improvements are significant. We will update these results in the revision.
>
> **Question 3: The paper does not mention some competing "Robust GNN" baselines.**
>
> Answer: (1) In practice, for a given graph, we do not have the prior knowledge about if the graph is clean, has noisy features or has adversarial features. Therefore, algorithms like the proposed AirGNN that can work under multiple abnormal feature settings are desired. However, these GNN defense models have been specifically designed for the adversarial setting. Thus choosing them as the baselines and evaluating them under multiple settings seem unfair to them. That is the major reason we exclude them in the current version.
>
> (2) As suggested by the reviewer, including these defense methods and showing the performance comparison can further demonstrate the advantages of the proposed AirGNN. Thus, we follow the suggestion and conduct experiments to include several representative robust models, including Robust GCN, GCN-Jaccard, and GCN-SVD. The comparison in the noise setting on Cora dataset is shown in the following table:
>
> | Ratio of noisy nodes | 1% | 2% | 3% |  4% |  5% |  8% |  10% |  15% | 20% |  25% | 30% |
> | - | - | - | - | - | - | - | - | - | - | - | - |
> | **Abnormal nodes** |
> | RGCN | 0.497 | 0.473 | 0.479 | 0.360 | 0.268 | 0.335 | 0.236 | 0.272 | 0.204 | 0.215 | 0.201 |
> | GCN-SVD | 0.200 | 0.200 | 0.150 | 0.238 | 0.160 | 0.250 | 0.170 | 0.223 | 0.157 | 0.144 | 0.150 |
> | GCN-Jaccard | 0.040 | 0.220 | 0.193 | 0.118 | 0.136 | 0.189 | 0.143 | 0.152 | 0.150 | 0.137 | 0.127 |
> | AirGNN | 0.650 | 0.511 | 0.485 | 0.502 | 0.496 | 0.421 | 0.463 | 0.392 | 0.363 | 0.396 | 0.343 |
> | **Normal nodes** |
> | RGCN | 0.800 | 0.767 | 0.727 | 0.679 | 0.636 | 0.569 | 0.500 | 0.410 | 0.330 | 0.290 | 0.251 |
> | GCN-SVD | 0.674 | 0.621 | 0.537 | 0.458 | 0.387 | 0.322 | 0.292 | 0.267 | 0.253 | 0.217 | 0.197
> | GCN-Jaccard | 0.762 | 0.742 | 0.716 | 0.695 | 0.664 | 0.617 | 0.560 | 0.466 | 0.399 | 0.339 | 0.261 |
> | AirGNN | 0.834 | 0.809 | 0.804 | 0.804 | 0.791 | 0.774 | 0.721 | 0.690 | 0.626 | 0.601 | 0.565 |
>
> The comparison in the adversarial setting on Cora dataset is shown in the following table:
>
> | Perturbation budget | 1  |  2  |  5  |  10  |  20  |  50  |  80 |
> | - | - | - | - | - | - | - | - |
> | RGCN | 0.800 | 0.775 | 0.717 | 0.633 | 0.558 | 0.477 | 0.458 |
> | GCN-SVD | 0.742 | 0.717 | 0.642 | 0.567 | 0.442 | 0.383 | 0.325 |
> | GCN-Jaccard | 0.775 | 0.717 | 0.633 | 0.517 | 0.317 | 0.133 | 0.125 |
> | AirGNN | 0.850 | 0.832 | 0.812 | 0.800 | 0.738 | 0.680 | 0.650 |
>
> The results from the tables highlight one big advantage of AirGNN: AirGNN does not assume the type of abnormal features, and it can work significantly better than these robust models under different types of abnormal features.
>
> (3) Another big advantage of AirGNN is that it does not sacrifice the performance when the dataset is clean or does not contain abnormal features. In other words, AirGNN is intrinsically robust by design. This can be verified in the comparison with APPNP on clean datasets:
>
> | Dataset | Cora | CiteSeer | PubMed | Photo | Physics |
> | :-: | :-: | :-: | :-: | :-: | :-: |
> | APPNP | 0.842 | 0.719 | 0.804 | 0.901 | 0.936 |
> | AirGNN | 0.839 | 0.726 | 0.806 | 0.906 | 0.936 |
>
> In contrast, in order to improve robustness, those robust models will sacrifice performance when the dataset is clean as showed in their original papers. which makes them less practical.
>
> These results clearly justify the advantages of AirGNN. We will extensively include more related works and results in the revision.
>
> **Question 4: The authors do not give a section that discusses the limitations and potential negative society impact.**
>
> **Answer**: We will include the discussion on the limitation and potential negative society in the revision.
>
> **Question 5: Not sure if it is a novel method.**
>
> **Answer**: As verified and discussed in the above responses, AirGNN is intrinsically robust by design. This is due to the principally designed adaptive message passing (AMP) which can adaptively reduce the impact of abnormal features. The model significantly improves the performance on datasets with various kinds of abnormal features (e.g., complicated noise settings and adversarial settings), without hurting the performance on clean datasets. The model design is effective, efficient, and reliable to deploy in various kinds of unknown environments where it is unclear whether the datasets are clean, adversarial, or noisy.  To the best of our knowledge, AirGNN is the first GNN algorithm that is intrinsically robust to abnormal features by design and can be applied to various kinds of environments which make it very practical in deployment.
>
> Overall, we hope that we have addressed the concerns in your comments, and please kindly let us know if there is any further concern, and we are happy to clarify.

---

> > ### Comment · Reviewer_tp11 · 2021-08-30
> > **Response**
> >
> > Thanks for the reply.
> >
> > I read reviewers' comments as well as the authors' replies. In general, the authors provide enough additional experiments to stress my concern. However, I still think it is not perfect to add so many additional experiments result in the rebuttal stage instead of showing them in the initial submission.
> >
> > I will raise my score to 6.

---

> > > ### Author Response · Authors · 2021-08-31
> > > **Thanks for the positive feedback**
> > >
> > > Dear reviewer,
> > >
> > > Thank you for your positive feedback. We will incorporate the additional results in the revision.
> > >
> > > Authors

---

### Official Review · Reviewer_Sbcq · 2021-07-16

**Rating:** 6
**Confidence:** 4

**Summary:**

This paper targets to design GNNs with stronger resilience to abnormal node features. Empirical examination is firstly conducted by replacing the features of randomly selected nodes with random Gaussian noise. The comparison between models with and without the residual connections show the helpfulness of residual connections on learning with abnormal features.

**Main Review:**

Although the designed model is simply reasonable, the missing of key related work in discussion and in evaluation comparison makes it hard to know how the proposed solution is effective.  The related work discussion is short and missing the closely related work that also addresses abnormal or noisy feature problems in GNN. The baselines include only the representative GNNs, not taking into account the related work of other robust GNN models.
For example:
Robust Graph Convolutional Networks Against Adversarial Attacks  in KDD 2019
GNNGuard: Defending Graph Neural Networks against Adversarial Attacks in NeurIPS 2020
And many other GNN defense models listed in Table 3 in paper entitled “Adversarial Attack and Defense on Graph Data: A Survey”.


**Time Spent Reviewing:**

5 hours

---

> ### Author Response · Authors · 2021-08-10
> **Response to all concerns by Reviewer Sbcq**
>
> Dear reviewer,
>
> Thanks for the valuable feedback. Here we provide detailed responses to address the concerns in your comments, and we will carefully incorporate them in the revision.
>
> Question: The missing of key related work in discussion and in evaluation comparison makes it hard to know how the proposed solution if effective. The related work is short. The baselines do not include other robust GNN models, such as RobustGNN, GNNGuard and other GNN defense models listed in the paper "Adversarial Attack and Defense on Graph Data: A Survey".
>
> Answer: (1) In practice, for a given graph, we do not have the prior knowledge about if the graph is clean, has noisy features or has adversarial features. Therefore, algorithms like the proposed AirGNN that can work under multiple abnormal feature settings are desired. However, these GNN defense models have been specifically designed for the adversarial setting. Thus choosing them as the baselines and evaluating them under multiple settings seem unfair to them. That is the major reason we exclude them in the current version.
>
> (2) As suggested by the reviewer, including these defense methods and showing the performance comparison can further demonstrate the advantages of the proposed AirGNN. Thus, we follow the suggestion and conduct experiments to include several representative robust models, including Robust GCN, GCN-Jaccard, and GCN-SVD. The comparison in the noise setting on Cora dataset is shown in the following table:
>
> | Ratio of noisy nodes | 1% | 2% | 3% |  4% |  5% |  8% |  10% |  15% | 20% |  25% | 30% |
> | - | - | - | - | - | - | - | - | - | - | - | - |
> | **Abnormal nodes** |
> | RGCN | 0.497 | 0.473 | 0.479 | 0.360 | 0.268 | 0.335 | 0.236 | 0.272 | 0.204 | 0.215 | 0.201 |
> | GCN-SVD | 0.200 | 0.200 | 0.150 | 0.238 | 0.160 | 0.250 | 0.170 | 0.223 | 0.157 | 0.144 | 0.150 |
> | GCN-Jaccard | 0.040 | 0.220 | 0.193 | 0.118 | 0.136 | 0.189 | 0.143 | 0.152 | 0.150 | 0.137 | 0.127 |
> | AirGNN | 0.650 | 0.511 | 0.485 | 0.502 | 0.496 | 0.421 | 0.463 | 0.392 | 0.363 | 0.396 | 0.343 |
> | **Normal nodes** |
> | RGCN | 0.800 | 0.767 | 0.727 | 0.679 | 0.636 | 0.569 | 0.500 | 0.410 | 0.330 | 0.290 | 0.251 |
> | GCN-SVD | 0.674 | 0.621 | 0.537 | 0.458 | 0.387 | 0.322 | 0.292 | 0.267 | 0.253 | 0.217 | 0.197
> | GCN-Jaccard | 0.762 | 0.742 | 0.716 | 0.695 | 0.664 | 0.617 | 0.560 | 0.466 | 0.399 | 0.339 | 0.261 |
> | AirGNN | 0.834 | 0.809 | 0.804 | 0.804 | 0.791 | 0.774 | 0.721 | 0.690 | 0.626 | 0.601 | 0.565 |
>
> The comparison in the adversarial setting on Cora dataset is shown in the following table:
>
> | Perturbation budget | 1  |  2  |  5  |  10  |  20  |  50  |  80 |
> | - | - | - | - | - | - | - | - |
> | RGCN | 0.800 | 0.775 | 0.717 | 0.633 | 0.558 | 0.477 | 0.458 |
> | GCN-SVD | 0.742 | 0.717 | 0.642 | 0.567 | 0.442 | 0.383 | 0.325 |
> | GCN-Jaccard | 0.775 | 0.717 | 0.633 | 0.517 | 0.317 | 0.133 | 0.125 |
> | AirGNN | 0.850 | 0.832 | 0.812 | 0.800 | 0.738 | 0.680 | 0.650 |
>
> The results from the tables highlight one big advantage of AirGNN: AirGNN does not assume the type of abnormal features, and it can work significantly better than these robust models under different types of abnormal features.
>
> (3) Another big advantage of AirGNN is that it does not sacrifice the performance when the dataset is clean or does not contain abnormal features. In other words, AirGNN is intrinsically robust by design. This can be verified in the comparison with APPNP on clean datasets:
>
> | Dataset | Cora | CiteSeer | PubMed | Photo | Physics |
> | :-: | :-: | :-: | :-: | :-: | :-: |
> | APPNP | 0.842 | 0.719 | 0.804 | 0.901 | 0.936 |
> | AirGNN | 0.839 | 0.726 | 0.806 | 0.906 | 0.936 |
>
> In contrast, in order to improve robustness, those robust models will sacrifice performance when the dataset is clean as showed in their original papers. which makes them less practical.
>
> These results clearly justify the advantages of AirGNN. We will extensively include more related works and results in the revision.
>
> Overall, we believe that we have addressed the concerns in your comments, and we appreciate it if you could consider adjusting the score if our responses are helpful. Please kindly let us know if there is any further concern, and we are happy to clarify.

---

> > ### Comment · Reviewer_Sbcq · 2021-08-27
> > **Re: response**
> >
> > Thank you for providing more results. I raised my rating to 6.

---

> > > ### Author Response · Authors · 2021-08-29
> > > **Thank you for the positive feedback.**
> > >
> > > Dear reviewer,
> > >
> > > Thank you for the positive feedback. We will carefully incorporate what you suggest in the revision of this paper.
> > >
> > > Authors

---

### Official Review · Reviewer_TraF · 2021-07-17

**Rating:** 5
**Confidence:** 3

**Summary:**


This paper observed interesting phenomenons: feature aggregation helps smooth out abnormal features but would lead to over-smoothing for normal features, residual link helps adjust feature smoothness for normal features but may hurt the performance when we have abnormal features. Therefore, it points out that we need to tradeoff between normal and abnormal features while designing GNNs. Then, based on the observation, this work proposed the AMP method and the AirGNN model, which are shown by experiments to be effective under various abnormal feature scenarios.



**Ethical Concerns:**

N/A.

**Limitations And Societal Impact:**


The limitation of this work is not discussed in the paper.


**Main Review:**


This paper provides very interesting observations and finds the tradeoff problem between normal and abnormal feature scenarios, the observations and the understandings (in section 2.2) make sense to me, and the results show in figure 5, figure 12(a)-16(a), are impressive.

However, I have the following questions/concerns:

1. What is the training objective function? Is it just Eq.(7) or Eq.(7) +classification loss?

2. Cora, Citeseer, and Pubmed are very classic datasets but they are not enough. It would be better if the authors can provide experimental results on other datasets in the main paper, instead of providing them in the appendix.

3. The author claimed that, in real-world applications, node features in graphs could often be abnormal such as being naturally noisy, then could you please provide some experimental results on some abnormal real-world datasets? For now, it seems to me that the authors are manually making the features in the rea-world dataset abnormal by replacing the raw features with Gaussian-generated features, so it is more like a synthetic scenario instead of a real-world scenario.

4. For figure 6, since they are on normal features, then why would the x-axis be "Ratio of Noisy Nodes(%)"? Why would the "Ratio of Noisy Nodes" influence the performance of GNNs with normal features? According to my understanding, noisy feature = abnormal features, am I misunderstanding?

5. I am curious, now you're using one Gaussian distribution to generate noisy features, what if the noise is generated in a more complicated way? (for example, what if different node class has different Gaussian distribution generation?)

6. I think the writing is not clear enough for me.




**Time Spent Reviewing:**

2 hrs

---

> ### Author Response · Authors · 2021-08-10
> **Response to all concerns by Reviewer TraF**
>
> Dear reviewer,
>
> Thanks for the recognition of our interesting observations and the impressive improvements by our proposed method.
> Here we provide detailed responses to address all concerns in your valuable feedback, and we will carefully incorporate them in the revision.
>
> **Question 1: What is the training objective function? Is it just Eq.(7) or Eq.(7) +classification loss?**
>
> **Answer**: Eq. (7) is only used to derive the adaptive message passing (AMP) showed in Figure 4. Figure 3 further illustrates how AMP works in the k+1 layer. Thus, it is a building block that can be used in any GNN architectures to improve the resilience to abnormal features. In our work, we choose the decoupled architecture like APPNP as shown in Eqs 14 and 15. In other words, we first use Eq 14 to transform the original input features and then perform K steps of AMP. Similar to the majority of existing GNN models, the training objective is the cross-entropy classification loss on the labeled nodes with features from the K-th steps of AMP. The whole framework is trained on an end-to-end way. We will make this clear in the revision.
>
>
> **Question 2: It would be better to provide experimental results on other datasets in the main paper, instead of in the appendix.**
>
> **Answer**: We will move the experiment results on other datasets from the appendix to the main paper as much as we can.
>
> **Question 3: For figure 6, since they are on normal features, then why would the x-axis be "Ratio of Noisy Nodes"? Why would the "Ratio of Noisy Nodes" influence the performance of GNNs with normal features?**
>
> **Answer**: (1) In this paper, we consider the setting where a subset of nodes in the graph contain abnormal features, while the remaining nodes have normal features, as stated in the preliminary (line 70-72). Therefore, the abnormal features (i.e., features on abnormal nodes) will impact normal features (i.e., features on normal nodes) through feature propagation over the graph. This is why when the ratio of noisy nodes becomes larger, the performance on normal nodes (i.e., normal features) also decreases.
>
> (2) We believe the confusion comes from the concepts of normal/abnormal features (they are clarified in line 72). In fact, "normal features" in the title of Figure 6 means the performance on normal nodes (a subset of all nodes in the graph). We will make this clear in the revision.
>
>  **Question 4: The authors are manually making the features in the real-world datasets abnormal by replacing the raw features with Gaussian-generated features, so it is more like a synthetic scenario. Could you please provide some experimental results on some abnormal real-world datasets? What if the noise is generated in a more complicated way? For example, what if different node class has different Gaussian distribution generation?**
>
> **Answer**: (1) In this paper, we include two abnormal feature settings to evaluate the effectiveness of AirGNN, i.e., the noise feature setting and the adversarial feature setting. Given that the existing benchmark datasets only provide clean features, we simulate noise features by adding Gaussian noise in the benchmark datasets. However, the adversarial feature setting includes attacked benchmark datasets that can exist in real-world applications. In Section 4.4, we further analyze how adaptive residual in AirGNN helps remove the impact of abnormal features, and the same observations can be made for both noise feature and adversarial feature settings. Thus, these two settings (one is synthetic and one is real-world) provide reasonable test-beds to evaluate the effectiveness of AirGNN.
>
> (2) We also agree that in the real-world setting, the noise feature setting is more complicated than just a unified Gaussian noise. Thus, we follow the ways you suggest to generate more complicated noise in this setting. The first way is to add the zero-mean Gaussian distribution under different standard deviations, and the results on Cora dataset with 10\% noise nodes are showed in the following table:
>
> | Std of noise | 0.1 | 0.5 | 1.0 | 1.5 |
> | - | - | - | - | - |
> | **Abnormal nodes** |
> |GCN| 0.414 | 0.204 | 0.125 | 0.167 |
> |GAT| 0.348 | 0.123 | 0.114 | 0.117 |
> |APPNP| 0.442 | 0.203 | 0.156 | 0.156 |
> |AirGNN| 0.646 | 0.525 | 0.463 | 0.328 |
> | **Normal nodes** |
> |GCN| 0.788 | 0.721 | 0.615 | 0.628 |
> |GAT| 0.798 | 0.593 | 0.490 | 0.508 |
> |APPNP| 0.831 | 0.777 | 0.687 | 0.674 |
> |AirGNN| 0.831 | 0.795 | 0.721 | 0.694 |
>
>
> The second way is to add the noise under different Gaussian distributions for different nodes. Specifically, we synthesize random noise from three Gaussian distributions with different standard deviations (1,2 and 3), and randomly add these noises to the selected node.  The performance on Cora dataset is shown in the following table:
>
> | Ratio of noisy nodes | 1% | 2% | 3% |  4% |  5% |  8% |  10% |  15% | 20% |  25% | 30% |
> | :-: | :-: | :-: | :-: | :-: | :-: | :-: | :-: | :-: | :-: | :-: | :-: |
> | **Abnormal nodes** |
> | GCN | 0.140 | 0.170 | 0.100 | 0.120 | 0.138 | 0.146 | 0.104 | 0.162 | 0.110 | 0.170 | 0.143 |
> | GAT | 0.04 | 0.170 | 0.133 | 0.102 | 0.156 | 0.184 | 0.115 | 0.127 | 0.104 | 0.103 | 0.134 |
> | APPNP | 0.07 | 0.215 | 0.090 | 0.110 | 0.226 | 0.224 | 0.133 | 0.169 | 0.156 | 0.168 | 0.174 |
> | AirGNN | 0.640 | 0.455 | 0.427 | 0.485 | 0.364 | 0.419 | 0.292 | 0.305 | 0.315 | 0.272 | 0.258 |
> | **Normal nodes** |
> | GCN | 0.763 | 0.767 | 0.720 | 0.709 | 0.661 | 0.621 | 0.589 | 0.514 | 0.452 | 0.439 | 0.348 |
> | GAT | 0.714 | 0.720 | 0.670 | 0.664 | 0.602 | 0.529 | 0.518 | 0.412 | 0.371 | 0.337 | 0.336 |
> | APPNP | 0.811 | 0.776 | 0.750 | 0.733 | 0.759 | 0.689 | 0.608 | 0.527 | 0.486 | 0.453 | 0.440 |
> | AirGNN | 0.839 | 0.782 | 0.805 | 0.794 | 0.782 | 0.740 | 0.724 | 0.608 | 0.605 | 0.517 | 0.438 |
>
> From these results, we can observe that AirGNN outperforms others consistently in terms of performance on both abnormal nodes and normal nodes. Similar observations can be made on other datasets as well.  In summary, AirGNN can still work as expected even in more complicated noise settings.
>
>
> (3) We highlight that in Section 4.4, we carefully analyze how the adaptive residual in AirGNN helps remove the impact of abnormal features, and the same observations can be made for both noise feature and adversarial feature settings. In principle, the proposed AirGNN is intrinsically robust to all kinds of abnormal features, no matter it is noise, adversarial, missing, or others.
>
> **Question 5: The writing is not clear enough.**
>
> **Answer**: The confusion could majorly come from the concepts of abnormal/normal features and abnormal/normal nodes as well as the experimental settings, as clarified in the answer to Question 3. We will clarify these confusions and further proofread the paper in the revision.
>
>
> **Question 6: The limitation of this work is not discussed in the paper.**
>
> **Answer**: We will include a discussion on the limitation and society impact in the revision.
>
> Overall, we hope that we have addressed all concerns in your comments. Please kindly let us know if there is any further concern, and we are happy to clarify.

---

### Decision · Program_Chairs · 2021-09-27

**Decision:**

Accept (Poster)

**Comment:**

The paper discovers the phenomenon that residual connections can amplify GNNs' vulnerability against abnormal node features. It analyzes possible reasons and based on the understanding designs an effective message passing scheme to help solve the issue.

The observed phenomenon is interesting and important for related applications, and the study and design are reasonable with impressive results. The good response from the authors has clarified several concerns of the reviewers (better presentation, more thorough experiments, comparison with related work etc). The clarification in the response should be incorporated into the final version if accepted.